# INFODET: A DATASET FOR INFOGRAPHIC ELEMENT DETECTION

**Jiangning Zhu**[1], **Yuxing Zhou**[1], **Zheng Wang**[1], **Juntao Yao**[1]
**Yima Gu**[1], **Yuhui Yuan**[2] , **Shixia Liu**[1]*
[1]BNRist, Tsinghua University [2]Canva CORE

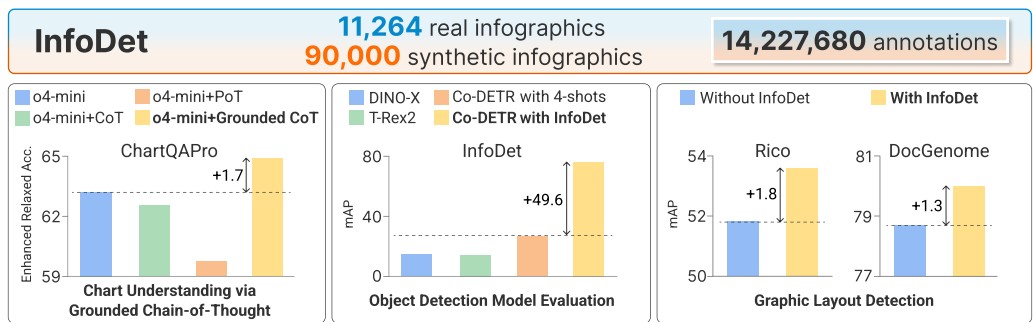

Figure 1: Key contributions: 1) An open-source dataset InfoDet. 2) Improvements on chart understanding, infographic element detection, and graphic layout detection.

## ABSTRACT

Given the central role of charts in scientific, business, and communication contexts, enhancing the chart understanding capabilities of vision-language models (VLMs) has become increasingly critical. A key limitation of existing VLMs lies in their inaccurate visual grounding of infographic elements, including charts and human-recognizable objects (HROs) such as icons and images. However, chart understanding often requires identifying relevant elements and reasoning over them. To address this limitation, we introduce InfoDet, a dataset designed to support the development of accurate object detection models for charts and HROs in infographics. It contains $11,264$ real and $90,000$ synthetic infographics, with over 14 million bounding box annotations. These annotations are created by combining the model-in-the-loop and programmatic methods. We demonstrate the usefulness of InfoDet through three applications: 1) constructing a Thinking-with-Boxes scheme to boost the chart understanding performance of VLMs, 2) comparing existing object detection models, and 3) applying the developed detection model to document layout and UI element detection.

 **Code:** https://github.com/InfoDet2025/InfoDet
 **Data & Dataset Card:** https://huggingface.co/datasets/InfoDet/InfoDet

## 1 INTRODUCTION

Charts are a fundamental medium for conveying data-driven insights across scientific, business, and communication domains. Consequently, improving vision-language models (VLMs) for chart understanding has become increasingly critical, driving significant advances in understanding plain charts (Huang et al., 2024)–minimal combinations of texts and charts. In practice, however, charts are often combined with icons and images of real-world objects, known as human-recognizable objects (HROs) (Borkin et al., 2016), to create infographics. By thoughtfully arranging texts, charts, and HROs, infographics transform abstract data into accessible insights through engaging visual

---

*Corresponding author.

designs. Due to their widespread use, automated infographic understanding using VLMs has become increasingly important for real-world applications such as news analysis (Chen & Chang, 2023) and accessible content generation (Hohenwalde & Hazim, 2025). However, previous studies (Masry et al., 2025; Li et al., 2025a) have identified a key limitation of existing VLMs: inaccurate visual grounding of infographic elements, which hinders the ability to associate the elements with the underlying data. This highlights the need for robust object detection models to support visual grounding and enhance chart understanding. Although considerable progress has been made in text detection (Long et al., 2021; Du et al., 2020), detecting charts and HROs—key elements linking abstract data to human perception—remains relatively underexplored.

Compared to natural scenes, element detection in infographics presents challenges for two reasons. First, infographic elements exhibit high intra-class variance. Charts vary widely in type, layout, and visual design, and HROs appear in diverse styles, spanning from realistic depictions to abstract representations of real-world objects. Second, the visual interplay between charts and HROs often results in ambiguous boundaries, making it difficult to distinguish one element from another in context. To effectively handle the highly varied infographic elements with ambiguous boundaries, the detection model needs to learn from a diverse set of infographics with accurate annotations. Existing datasets, however, primarily focus on plain charts without HROs (Battle et al., 2018; Deng et al., 2023), failing to capture the complexity of infographics. Borkin et al. (2016) have taken the first step in building a dataset with rich annotations, but their dataset is limited in scale, comprising only 393 samples due to the labor-intensive manual annotation process. To advance element detection in infographics, a large-scale dataset of diverse infographics with comprehensive annotations is required.

To fill this gap, we introduce InfoDet, a dataset for infographic element detection. It comprises a diverse collection of infographics from two sources: 1) real infographics collected from 10 online platforms, such as Visual Capitalist and Statista, and 2) synthetic infographics programmatically created from $1,072$ design templates. To effectively annotate the infographics, we combine programmatic and model-in-the-loop (Kirillov et al., 2023) methods. For the synthetic infographics, we programmatically derive the bounding boxes. For the real infographics, we co-develop an object detection model and the annotation process, allowing the model and the annotations to iteratively enhance each other. Specifically, we use the annotated synthetic infographics to fine-tune an InternImage-based object detection model (Wang et al., 2023), which is then employed to generate annotations for all real infographics. The generated annotations are reviewed and corrected by the experts through multiple rounds of refinement. In each round, expert feedback is utilized to enhance the annotation quality and refine the model, thereby progressively improving its accuracy. In total, InfoDet contains **11,264** real and **90,000** synthetic infographics, along with **14,227,680** bounding box annotations of texts, charts, HROs, and finer-grained sub-elements such as bars, axes, and legends. Table 6 in the Appendix provides a statistical comparison of InfoDet with existing chart datasets.

We demonstrate the usefulness of InfoDet through three applications (Fig. 1). First, we propose a Thinking-with-Boxes scheme that performs grounded chain-of-thought reasoning over elements. This grounded reasoning considerably improves the performance of OpenAI o4-mini on the challenging ChartQAPro benchmark (Masry et al., 2025). Second, we compare the performance of the state-of-the-art object detection models. The results show that the best-performing foundation models for object detection (*e.g.*, DINO-X (Ren et al., 2024)) still struggle to accurately detect infographic elements, whereas fine-tuning traditional object detection models (*e.g.*, Faster R-CNN (Ren et al., 2015)) with InfoDet achieves improved performance. These findings highlight the importance of sufficient high-quality training data for chart and HRO detection. Third, we apply our InternImage-based object detection model to out-of-domain graphic layout detection tasks, including document layout and UI element detection, demonstrating its generalization capability across broader domains.

The main contributions of this work are threefold:

- A large-scale dataset for infographic element detection with $101,264$ annotated infographics.

- An InternImage-based model for detecting charts and HROs in infographics.

- Three applications that show InfoDet's usefulness in chart understanding and object detection.

## 2 RELATED WORK

Based on the presence of HROs, chart datasets with element annotations can be classified into two categories: datasets of plain charts and datasets of infographic charts.

**Plain charts** present data in a minimal manner using texts and charts. Some datasets consist of programmatically created charts. FigureQA (Kahou et al., 2018) comprises $100,000$ charts created from randomly generated data using Bokeh. However, relying on randomly generated data limits real-world representativeness. To address this, Methani et al. (2020) use crawled data to create $224,377$ charts by randomly combining design parameters such as marker and line styles. Other datasets are constructed by collecting charts from existing literature or online platforms. VG-DCU (Dou et al., 2024) consists of $15,197$ SVG-based charts, from which bounding box annotations are extracted by analyzing the SVG elements. VisImages (Deng et al., 2023) is constructed by gathering $12,267$ images from IEEE VIS publications and manually annotating $35,096$ charts within them. While these datasets facilitate object detection model training for plain charts, such models often struggle with the widely used infographics, where diverse HROs and their interplay with the charts introduce significant variability.

To better support the analysis of **infographic** designs, Borkin et al. (2016) pioneered the creation of an infographic dataset with rich annotations. They utilize an existing database of real infographics and manually annotate the polygons of their elements. However, this dataset is limited in scale, comprising only 393 samples due to the labor-intensive manual annotation process. As a result, this dataset is unsuitable for training object detection models that require strong generalization. In contrast, InfoDet combines a model-in-the-loop annotation method for real infographics and a programmatic annotation method for synthetic infographics, resulting in $101,264$ annotated infographics that effectively support object detection model development.

## 3 INFODET CONSTRUCTION METHOD

Fig. 2 provides an overview of the dataset construction pipeline, which includes two main steps: infographic collection and infographic annotation.

### 3.1 INFOGRAPHIC COLLECTION

Previous studies (Zhu-Tian et al., 2020; Zhu et al., 2025) have highlighted the complementary benefits of real and synthetic data: the former captures authentic design practices, while the latter offers controlled variation and scalability for robust training and evaluation. Informed by this finding,

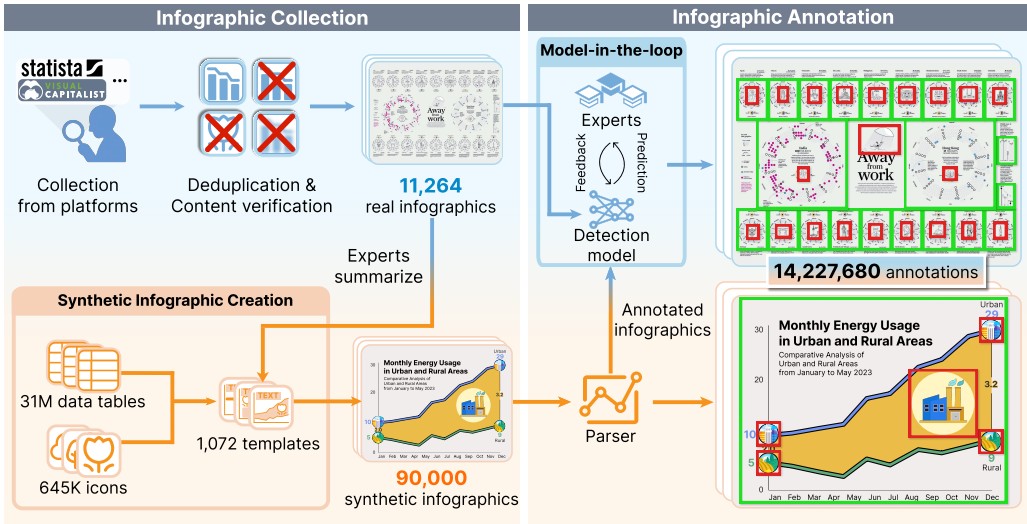

Figure 2: The construction pipeline for the InfoDet dataset.

we collect infographics from two sources to balance authenticity, diversity, and scalability: 1) real infographics from chart-rich online platforms, and 2) synthetic infographics created programmatically.

**Real infographic collection** We collect infographics from 10 chart-rich online platforms that permit research use, such as Visual Capitalist and Statista. The complete list is provided in Appendix B. To enhance the quality of the infographics, we implement a two-step filtering process: deduplication and content verification. In deduplication, we remove infographics that exhibit high CLIP similarity (Radford et al., 2021) ($\geq 0.9$) and low perceptual hashing distance (Jain et al., 2019) ($\leq 2$) relative to other infographics. In content verification, we prompt GPT-4o mini to identify and remove images that are blurry, lack charts or HROs, are photographs instead of graphic designs, or contain personal or sensitive content. After filtering, the collection is refined to $11,264$ high-quality infographics.

**Synthetic infographic creation** We employ a template-based method (Li et al., 2025b) to create synthetic infographics. This method utilizes $1,072$ design templates derived from representative real infographics. Each template specifies: 1) the presence and relative positions of charts, texts, and HROs, and 2) the type and visual style of the charts. An infographic is created by filling the template with: 1) data tables for chart creation, 2) descriptive texts, and 3) selected HROs. To ensure diversity, we sample data tables from VizNet (Hu et al., 2019), a large-scale dataset containing over 31 million tables and associated metadata. Charts are created from the sampled data tables as specified by the template. Descriptive texts for the charts are generated using GPT-4o mini. HROs with the highest CLIP similarity to the descriptive texts are selected from the IconQA dataset (Lu et al., 2021), which contains over 645K icons. Using this template-based method, we generate $90,000$ synthetic infographics. Example templates and infographics generated from them are provided in Appendix C.

## 3.2 Infographic Annotation

Given the differences in collecting real and synthetic infographics, we adopt two annotation methods: a programmatic method for synthetic infographics and a model-in-the-loop method for real ones.

**Programmatic synthetic infographic annotation** Synthetic infographic annotations are programmatically generated with a parser integrated into the infographic generation process. This parser extracts bounding boxes for texts, charts, and HROs from the corresponding SVG file, which encodes the visual and structural details of the infographic. Additionally, the parser leverages information from the design template to classify charts and HROs into subcategories: charts are categorized into 75 distinct types, while HROs are labeled as either data-related or theme-related objects. The complete list of chart types is provided in Appendix D.

**Model-in-the-loop real infographic annotation** To reduce human labor in the annotation, we aim to leverage object detection models for assistance. However, there is an absence of specialized detection models for charts and HROs. To address this, we employ a model-in-the-loop annotation method (Kirillov et al., 2023). This method co-develops an object detection model and the annotation process, allowing the model and the annotations to iteratively enhance each other. Specifically, using the annotated synthetic infographics, we build an object detection model by fine-tuning InternImage-L (Wang et al., 2023) along with the DINO (Zhang et al., 2023) detector. This fine-tuned model is then employed to generate annotations for all real infographics. However, since the synthetic infographics do not fully represent the diversity of all infographics, the fine-tuned object detection model is prone to errors. To mitigate this, we conduct multiple rounds of annotation refinement and model enhancement with the experts. In each round, the experts review and correct the auto-generated annotations, and the feedback is used to further fine-tune the model, progressively improving its accuracy. More details on the expert background and the refinement rounds are provided in Appendix E. At the end of the refinement process, we randomly sample $1,250$ infographics to evaluate the quality of the generated annotations. Results show that the generated annotations achieve a precision of $93.9\%$ and a recall of $96.7\%$, comparable to those of widely used object detection datasets, such as COCO (Lin et al., 2014) ($83.0\%$ recall and $71.9\%$ precision) and Objects365 (Shao et al., 2019) ($92.0\%$ recall and $91.7\%$ precision), as reported by Shao et al. (2019). We analyze the remaining model errors and discuss the dataset scope in Appendix F.

## 3.3 Statistics and Dataset Analysis

InfoDet contains **101,264** infographics, including **11,264** real and **90,000** synthetic infographics. To complement the dataset with text annotations, we use the widely adopted OCR model PP-OCRv4 (Du

et al., 2020) to annotate all real infographics and extract text annotations from the generation process for synthetic infographics. For completeness, we also extract mark-level annotations of 26 categories of chart sub-elements, such as bars, axes, and legends. The list of sub-element categories is provided in Appendix G. In addition, we extract segmentation masks for the charts, HROs, and chart sub-elements using SAM (Kirillov et al., 2023) to support finer-grained analysis. Across these infographics, we annotate a total of **7,284,892** text elements (each corresponding to a line of text), **310,299** charts, **1,080,598** HROs, and **5,551,891** sub-elements. The detailed statistics are provided in Appendix D. Beyond the basic statistics, we have verified that our dataset exhibits high diversity, no harmful bias, and high fidelity of synthetic infographics (See Appendix H). To ensure consistent evaluation, we split InfoDet into a training set of $96,264$ infographics and a test set of $5,000$ infographics, while maintaining the same proportion of real and synthetic infographics in both sets.

## 4 EXPERIMENTS

In this section, we first construct a Thinking-with-Boxes scheme to enhance the performance of the latest VLMs. We then evaluate the performance of existing object detection models. Finally, we apply the InternImage-based object detection model to graphic layout detection tasks.

### 4.1 THINKING-WITH-BOXES VIA GROUNDED CHAIN-OF-THOUGHT

Latest VLMs, such as OpenAI's o3/o4-mini, demonstrate chain-of-thought reasoning capability with images through seamless image manipulations, including automatic zooming and cropping. Considering that chart understanding requires more complex, fine-grained visual reasoning over the elements within infographic images (Lin et al., 2025), we construct a Thinking-with-Boxes scheme to enhance VLMs by explicitly providing grounded annotations of texts, charts, and HROs along with additional layered infographic images. The bounding boxes are predicted using our infographic-oriented object detection model and an OCR model. With this scheme, we prompt the VLMs to output reasoning trajectories over the grounded regions, referred to as *grounded chain-of-thought* (grounded CoT), which guide the model to think step-by-step before achieving the final answer.

#### 4.1.1 GROUNDED CHAIN-OF-THOUGHT PROMPTING

As shown in Fig. 3(B$_1$), we provide the VLM with detected elements in two modalities—visual prompts, by overlaying boxes on the infographic image, and textual descriptions of each element—to study the reasoning preferences of the evaluated VLMs.

For the **visual prompts**, we overlay bounding boxes on top of the infographics, each labeled with an alphabetical ID. To improve clarity, the bounding boxes are rendered in contrastive colors against the background, and the ID labels are placed to minimize overlap. However, even with these measures,

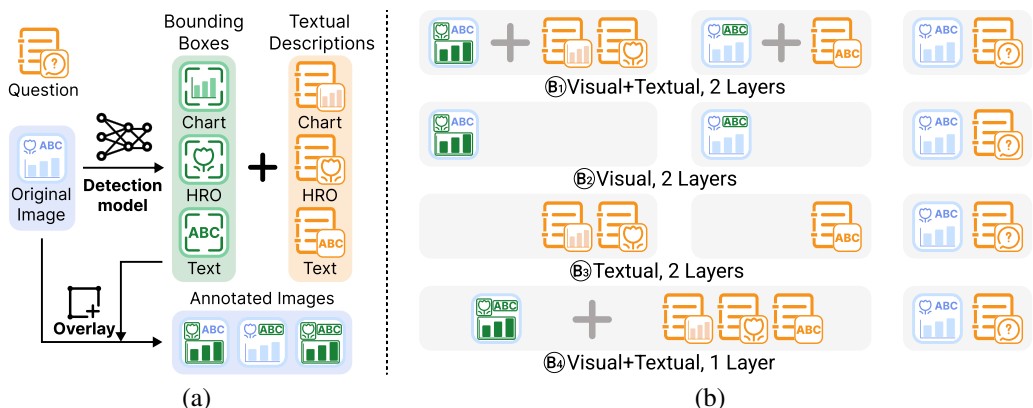

Figure 3: The Thinking-with-Boxes scheme: (a) the charts, HROs, and texts are detected and overlaid onto the original image to create annotated images with grounded elements; (b) the input of the grounded chain-of-thought method (B$_1$) and its ablated variants (B$_2$, B$_3$, B$_4$).

Table 1: Performance of o1, o3, and o4-mini with different prompting methods. The best one is **bold**.

| Chart Group | o1 | | | | o3 | | | | o4-mini | | | |
|---|---|---|---|---|---|---|---|---|---|---|---|---|
| | Direct | CoT | PoT | Grounded CoT (ours) | Direct | CoT | PoT | Grounded CoT (ours) | Direct | CoT | PoT | Grounded CoT (ours) |
| Plain, Single | 57.8 | 57.8 | 56.1 | **60.1** | 56.8 | **57.7** | 57.5 | 57.2 | 58.1 | 57.9 | 55.3 | **60.6** |
| Plain, Multiple | 63.7 | 65.1 | 62.2 | **65.4** | 62.8 | 61.0 | 58.8 | **63.4** | 66.7 | 66.1 | 62.3 | **66.9** |
| Infographic, Single | 66.4 | 64.3 | 60.9 | **67.8** | 64.9 | 59.5 | 64.2 | **67.7** | 67.4 | 64.4 | 67.5 | **68.4** |
| Infographic, Multiple | 66.0 | 67.6 | 66.8 | **71.9** | 66.0 | 64.9 | 64.2 | **68.8** | 70.6 | 69.2 | 64.7 | **72.5** |
| Overall | 61.4 | 61.9 | 60.0 | **64.1** | 60.6 | 60.0 | 59.5 | **61.6** | 63.2 | 62.5 | 59.7 | **64.9** |

overlap between bounding boxes remains inevitable in regions with dense texts and HROs. To mitigate this, we propose to separate the visual prompts into two layers: one containing charts and HROs, and the other containing texts. We also provide **textual description** of each element to ease the challenge of simultaneously locating and interpreting their content. Please refer to Appendix I.1 for detailed prompts and a comparison of visual prompts rendered in one versus two layers.

### 4.1.2 EXPERIMENTAL SETUP

We evaluate the chart understanding capability of VLMs using ChartQAPro (Masry et al., 2025), ChartQA (Masry et al., 2022), and PlotQA (Methani et al., 2020). To better analyze the performance of our method, we manually categorize them into four groups based on two criteria: whether the charts are **plain** or **infographic**, and whether there are **single** or **multiple** charts. We assess three state-of-the-art VLMs: OpenAI's o1, o3, and o4-mini. For each VLM, we compare our method against three widely used baseline prompting methods: 1) **Direct** prompting with the chart image and the question, 2) **Chain-of-Thought** (Wei et al., 2022) (CoT), which prompts the model to reason step-by-step for the provided image and question, and 3) **Program-of-Thought** (Chen et al., 2023) (PoT), which prompts the model to generate a Python code that prints the final answer. The performance is measured using the enhanced relaxed accuracy (Masry et al., 2025). Please refer to Appendix I.1 for the detailed prompts and the enhanced relaxed accuracy implementation.

### 4.1.3 RESULTS AND ANALYSIS

The ChartQAPro results are shown here. The ChartQA and PlotQA results are in Appendix J.1.

**Effectiveness of grounded CoT prompting**   As shown in Table 1, prompting the latest VLMs to think step-by-step or write Python code does not significantly improve their performance. We attribute this to the reasoning-centric design of the VLMs, which inherently reduces the dependence on explicit prompts for step-by-step reasoning. In contrast, our method enhances chart understanding performance by providing grounded infographic elements. In particular, our method performs comparably on plain, single charts and shows better performance on infographic charts and images with multiple charts, leading to improved overall performance. To assess the robustness of the observed gains, we include an additional set of runs on o3 and present the results in Appendix J.2. As shown in Fig. 4, the grounded annotations of elements effectively guide the VLM to reason step-by-step and arrive at the correct answer. We provide more examples in Appendix J.3. Despite its strong visual reasoning capability, o3 encounters instruction-following issues, resulting in slightly lower performance compared to o1 and o4-mini. The detailed evaluation results and analysis of this issue are provided in Appendix J.4.

**Ablation Study**   We report ablation studies on o1 examining different prompt modality designs, separating grounded annotations into two layers, and adding in-context examples. Additional ablations that vary the object detection and OCR models are provided in Appendix J.5.

*Prompt modality*. Table 2(a) shows that using only visual prompts (Fig. 3(B$_2$)) or textual descriptions (Fig. 3(B$_3$)) results in a performance drop compared to combining both. This highlights their complementary roles in grounding infographic elements and supporting VLMs in chart understanding.

*Prompt separation*. Table 2(b) shows that separating the prompts into two layers leads to better performance than providing them in one layer (Fig. 3(B$_4$)). This suggests that reducing overlap through separation facilitates the visual grounding of infographic elements and improves chart understanding.

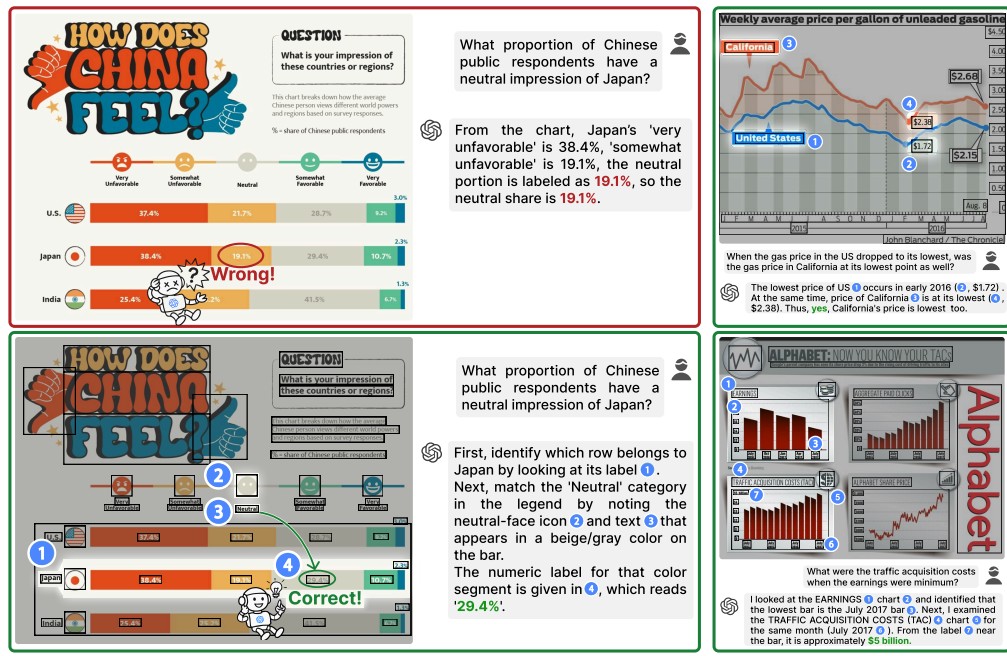

Figure 4: Grounded CoT guides the model to think step-by-step and achieve the correct answer.

Table 2: Ablation of the grounded CoT method. The best one is **bold**.

| (a) Prompt Modality | | | (b) Prompt Separation | | (c) In-Context Example | |
|---|---|---|---|---|---|---|
| Visual | Textual | Visual+Textual | 1-Layer | 2-Layer | With Example | Without Example |
| 62.8 | 61.6 | **64.1** | 62.3 | **64.1** | 61.5 | **64.1** |

*Incorporation of in-context examples*. Table 2(c) shows that incorporating in-context examples results in a performance drop. This indicates that the latest VLMs can perform reasoning tasks effectively without additional examples, which would instead introduce confusion and hinder performance.

## 4.2 EVALUATING OBJECT DETECTION MODELS

We evaluate the infographic element detection performance of 11 object detection models on InfoDet.

### 4.2.1 EXPERIMENTAL SETUP

**Models**   Existing object detection models can be classified into two categories: foundation models that support zero-/few-shot detection and traditional deep learning models that require fine-tuning before detecting novel classes. We select the representative models in each category, including seven foundation models (RegionCLIP (Zhong et al., 2022), Detic (Zhou et al., 2022), Grounding DINO (Liu et al., 2024), GLIP (Li et al., 2022b), MQ-GLIP (Xu et al., 2023), T-Rex2 (Jiang et al., 2024), and DINO-X (Ren et al., 2024)) and four traditional models (Faster R-CNN (Ren et al., 2015), YOLOv3 (Redmon & Farhadi, 2018), RTMDet (Lyu et al., 2022), and Co-DETR (Zong et al., 2023)).

**Evaluation protocol**   The above models are not tailored to detecting charts and HROs. To address this, we evaluate three adaptation methods: 1) **Zero-shot prompting**, which uses text prompts to define target classes, 2) **Few-shot prompting**, which uses $k$ randomly selected infographics to describe target classes, optionally augmented with text prompts, and 3) **Standard fine-tuning**, which updates model weights using annotated infographics, either with $k$ random example infographics or the entire InfoDet training set. We compare the models at two granularity levels: the element level targeting charts and HROs, and the mark level targeting sub-elements such as bars, axes, and legends. For both levels, the performance is measured using the average precision (AP) and recall

Table 3: Evaluation results of the foundation and the traditional models at the element level. The best one is **bold**.

(a) Zero-shot prompting

| Model | Average Precision (AP) | | Average Recall (AR) | |
|---|---|---|---|---|
| | Chart | HRO | Chart | HRO |
| RegionCLIP | 0.8 | 3.6 | 13.9 | 24.9 |
| Detic | 1.8 | 4.4 | 23.7 | 11.3 |
| Grounding DINO | 12.6 | 12.2 | **63.2** | **46.0** |
| GLIP | 13.5 | 11.2 | 44.9 | 33.2 |
| MQ-GLIP | 13.5 | 11.2 | 44.9 | 33.2 |
| DINO-X | **14.0** | **15.0** | 29.4 | 29.1 |

(b) Few-shot prompting, 4-shots

| Model | Average Precsion (AP) | | Average Recall (AR) | |
|---|---|---|---|---|
| | Chart | HRO | Chart | HRO |
| MQ-GLIP | **16.2** | 15.5 | **43.5** | **40.7** |
| T-Rex2 | 12.2 | **16.2** | 21.8 | 24.7 |

(c) Standard fine-tuning, 4-shots

| Model | Average Precsion (AP) | | Average Recall (AR) | |
|---|---|---|---|---|
| | Chart | HRO | Chart | HRO |
| RegionCLIP | 6.8 | 14.7 | 15.5 | 22.9 |
| Detic | 19.6 | 14.2 | 37.0 | 22.8 |
| Faster R-CNN | 3.4 | 1.0 | 10.8 | 1.5 |
| YOLOv3 | 5.5 | 4.0 | 16.2 | 13.1 |
| RTMDet | 12.8 | 18.9 | 44.2 | 49.1 |
| Co-DETR | **27.6** | **25.5** | **53.4** | **49.7** |

(d) Standard fine-tuning, InfoDet

| Model | Average Precsion (AP) | | Average Recall (AR) | |
|---|---|---|---|---|
| | Chart | HRO | Chart | HRO |
| RegionCLIP | 10.1 | 23.3 | 17.5 | 28.6 |
| Detic | 39.6 | 34.3 | 57.4 | 47.7 |
| Faster R-CNN | 78.9 | 49.0 | 80.8 | 52.7 |
| YOLOv3 | 14.7 | 25.5 | 43.2 | 35.7 |
| RTMDet | 83.7 | 53.6 | 86.4 | 59.9 |
| Co-DETR | **88.2** | **64.0** | **89.8** | **69.5** |

(AR) on the InfoDet test set. Please refer to Appendix I.2 for more details on text prompts, fine-tuning hyperparameters, and computational costs.

### 4.2.2 RESULTS AND ANALYSIS

**Comparing adaptation methods and object detection models at the element level** We evaluate all applicable adaptation methods for each model, except for standard fine-tuning, which is restricted to models that fit within the memory constraints of an NVIDIA Tesla V100 GPU. For few-shot prompting and fine-tuning methods, we use $k = 4, 10$, and $30$ randomly selected infographics. We average the results over 3 runs, excluding T-Rex2 and DINO-X due to their reliance on charged APIs. Table 3 shows the results for all models with $k = 4$. The full results, including the variance across runs, are available in Appendix K.1. We present our key findings as follows:

*Zero-shot and few-shot prompting exhibit limited performance*. Zero-shot prompting exhibits limited performance. As shown in Fig. 5(a), even state-of-the-art foundation models like DINO-X fail to interpret these concepts through textual prompts, often missing key components. Contrary to prior findings (Madan et al., 2024), providing annotated example infographics does not lead to notable improvements (Fig. 5(b)). We attribute this to the models' pretraining on natural scenes (Mathew et al., 2022), which provides limited exposure to graphic representations such as infographics. Consequently, the models lack the prior knowledge needed to effectively learn from the provided examples.

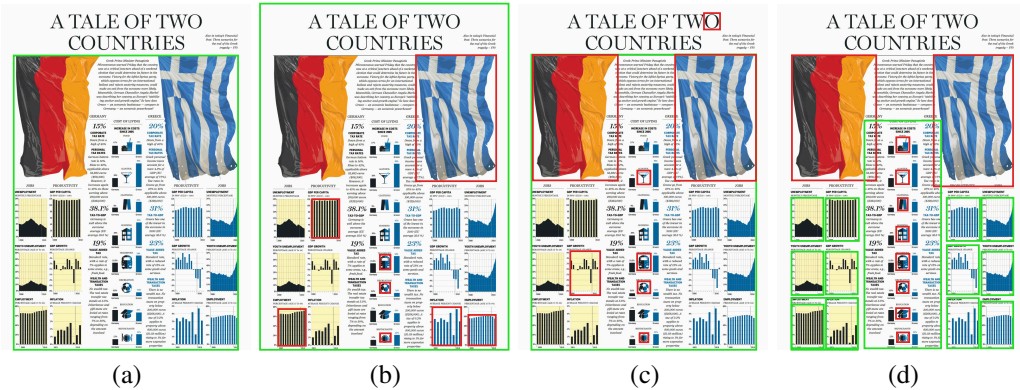

(a)         (b)         (c)         (d)

Figure 5: Detection results of evaluated object detection models: (a) zero-shot prompting with DINO-X; (b) 4-shot prompting with T-Rex2; (c) 4-shot fine-tuning with Co-DETR; (d) fine-tuning on InfoDet with Co-DETR. Bounding boxes in colors are the predictions for charts and HROs.

Table 4: Evaluation results of the traditional models at the element level. The best one is **bold**.

| Model | mAP | mAR |
|---|---|---|
| Faster R-CNN | $47.0 \pm 0.1$ | $49.9 \pm 0.1$ |
| YOLOv3 | $11.3 \pm 0.9$ | $21.5 \pm 0.5$ |
| RTMDet | $46.9 \pm 0.2$ | $59.5 \pm 0.5$ |
| Co-DETR | $\mathbf{69.6 \pm 0.4}$ | $\mathbf{77.0 \pm 0.3}$ |

*Standard fine-tuning improves performance*. Compared with zero-/few-shot prompting, fine-tuning with example infographics or the InfoDet training set yields higher performance. Few-shot experiments show that the performance improves significantly as the number of example infographics increases. Moreover, fine-tuning on InfoDet consistently outperforms few-shot fine-tuning, particularly for the traditional models. This is verified by Co-DETR's more accurate detection results after fine-tuning on InfoDet (Fig. 5(d)) compared to using 4 example infographics (Fig. 5(c)).

**Evaluating traditional models at the mark level** Due to the effectiveness of fine-tuning traditional models at the element level, we further evaluate them at the mark level. Table 4 shows the performance of each model averaged over 3 runs. Despite the increased difficulty of mark-level detection, which leads to a drop in mAP (*e.g.*, from 76.1% to 69.6% for Co-DETR), the models still perform effectively, supported by the large-scale training set of InfoDet. We also evaluate model generalizability via cross-dataset evaluation following Deng et al. (2023). Specifically, we fine-tune each model on either our infographics or VG-DCU (Dou et al., 2024) (plain charts with mark-level annotations) and test on the other. Models show a smaller drop in mAP when transferred from our infographics to VG-DCU than vice versa. For example, the mAP of Co-DETR drops 17.1% when transferring from our infographics to VG-DCU, compared with a much larger 53.7% drop in the opposite direction. This indicates that models trained on our infographics generalize more effectively due to the inclusion of infographic charts. The full experimental setup and results are provided in Appendix K.1.

## 4.3 APPLYING THE DEVELOPED MODEL TO GRAPHIC LAYOUT DETECTION

To demonstrate the broader applicability of InfoDet, we evaluate its effectiveness on graphic layout detection tasks by applying the InternImage-based model.

### 4.3.1 EXPERIMENTAL SETUP

We evaluate the InternImage-based model on three graphic layout detection datasets, Rico (Deka et al., 2017), DocGenome (Xia et al., 2024), and PosterLayout (Hsu et al., 2023). As shown in Table 5, we fine-tune five model variants for each dataset. Four of which are pre-trained on a different combination of ImageNet-22K (Deng et al., 2009), Objects365 (Shao et al., 2019), COCO (Lin et al., 2014), and InfoDet. The fifth variant is pre-trained on all four datasets and leverages a hierarchy-guided matching (HGM) scheme on InfoDet that encourages matched predictions to preserve the hierarchical relationships between charts and their sub-elements in the ground truth. Please refer to Appendix I.3 for more details on the graphic layout detection datasets, the hierarchy-guided matching scheme, fine-tuning hyperparameters, and computational costs.

### 4.3.2 RESULTS AND ANALYSIS

As shown in Table 5, pre-training on InfoDet improves model performance when fine-tuned on these datasets, both on its own and when combined with existing large-scale pre-training data. With the growing interest in integrating multiple datasets for training foundation models (Yang et al., 2024),

Table 5: Performance of the detection models with different pre-training. The best one is **bold**.

| Pre-Training | Rico | DocGenome | PosterLayout |
|---|---|---|---|
| - | 42.1 | 69.0 | 62.5 |
| InfoDet | 50.6 | 74.4 | 73.6 |
| ImageNet-22K, Objects365, COCO | 51.8 | 78.7 | 74.9 |
| ImageNet-22K, Objects365, COCO, InfoDet | 53.6 | 80.0 | 76.0 |
| ImageNet-22K, Objects365, COCO, InfoDet w. HGM | **54.4** | **80.8** | **76.4** |

InfoDet serves as a useful addition to existing resources for graphic layout detection. Additionally, incorporating HGM yields additional gains, suggesting that exploiting the hierarchical structure in InfoDet is a promising direction for future research.

## 5 CONCLUSION

In this paper, we introduce InfoDet, a dataset designed to support infographic element detection. It features a diverse collection of real and synthetic infographics, along with bounding box annotations for texts, charts, HROs, and finer-grained sub-elements. Three applications demonstrate that InfoDet is not only valuable for developing visual reasoning methods but also broadly applicable to tasks such as object detection model evaluation and graphic layout detection. Although InfoDet has proven effective, there remain promising directions for future work. For example, analyzing the annotated infographics to uncover design principles could advance automated infographic design.

## ETHICS STATEMENT

To ensure the integrity of this work, we carefully consider several ethical aspects when collecting real infographics from online platforms. First, we manually collect the real infographics and release only their source URLs on Hugging Face, without hosting or redistributing third-party content. We have reviewed the data usage policies of the platforms and confirmed that they either explicitly permit (*e.g.*, Statista, Visual Capitalist) or do not prohibit (*e.g.*, Daily Infographics, Infographics Archive) the use of their content for research purposes, including the sharing of source URLs. Details of the data usage policies and licenses of each platform are provided in Appendix B. Second, to exclude harmful and sensitive content from our dataset, we: 1) collect only from reputable public platforms, which generally filter such content; and 2) use GPT-4o mini to flag potentially harmful or sensitive infographics, which are then manually verified and removed. Finally, we release our dataset and model strictly for academic research purposes.

## ACKNOWLEDGMENTS

This work was supported by the National Natural Science Foundation of China under grant U21A20469. The authors would like to thank Duan Li for his valuable contributions to the discussions.

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

## A    STATISTICAL COMPARISON OF INFODET WITH EXISTING CHART DATASETS

Table 6 provides a statistical comparison of InfoDet with existing datasets. Unlike most existing datasets that focus on plain charts, InfoDet is specifically designed for infographic charts, where charts, texts, and HROs are tightly integrated in visually complex layouts. Compared to the dataset by Borkin et al. (2016), the only existing infographic dataset with mark-level annotations, InfoDet is significantly larger in scale and better suited for training object detection models.

Table 6: Statistics of existing chart datasets.

| Dataset | # Real | # Synthetic | Infographic? |
|---|---|---|---|
| InfoDet (ours) | 11, 264 | 90, 000 | ✓ |
| Borkin et al. (2016) | 393 | 0 | ✓ |
| FigureQA (Kahou et al., 2018) | 0 | 100, 000 | - |
| PlotQA (Methani et al., 2020) | 0 | 224, 377 | - |
| Beagle (Battle et al., 2018) | 41, 000 | 0 | - |
| VisImages (Deng et al., 2023) | 12, 267 | 0 | - |
| VG-DCU (Dou et al., 2024) | 4, 515 | 10, 682 | - |

## B    ONLINE PLATFORMS FOR REAL INFOGRAPHIC COLLECTION

We collect the real infographics from the 10 online platforms listed in Table 7. The infographic collection strictly adheres to the copyright and licensing regulations of the respective platforms.

Table 7: Infographic platforms and licenses.

| Platform | Website Link | Licenses |
|---|---|---|
| Statista | statista.com | CC BY-NC |
| Visual Capitalist | visualcapitalist.com | Customized: "For individuals and small organizations (5 people or less, or < $1 million in revenue), we allow you to use our visuals in a variety of use cases for free. These include personal and commercial use cases, such as: embedding our graphics in a newsletter, report, video, presentation, or on your website." |
| World Statistics | world-statistics.org | CC BY |
| Our World in Data | ourworldindata.org | CC BY |
| OECD | oecd.org | CC BY |
| Openverse | openverse.org | CC |
| The Conversation | theconversation.com | CC BY-ND |
| Kaiser Family Foundation | kff.org | CC BY-NC-ND |
| Daily Infographics | dailyinfographics.com | Customized. No prohibition on research use or sharing of source URLs. |
| Infographics Archive | infographicsarchive.com | Customized. No prohibition on research use or sharing of source URLs. |

## C    EXAMPLE SYNTHETIC INFOGRAPHICS

We employ a template-based method to create synthetic infographics. Fig. 6 shows examples of design templates and infographics generated from them.

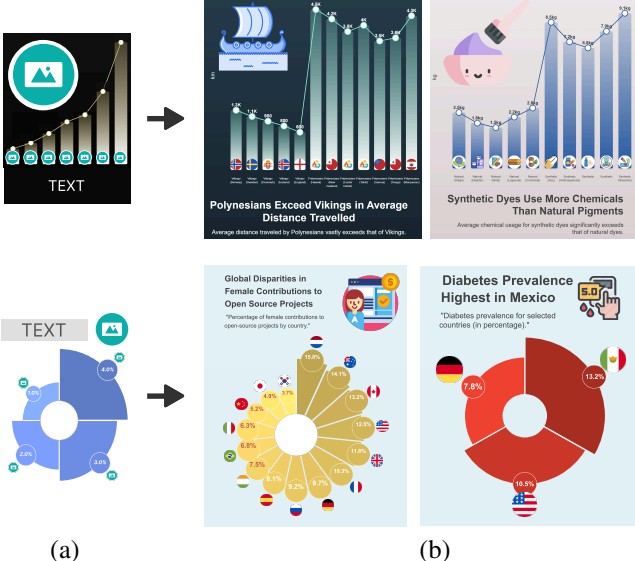

(a)          (b)

Figure 6: Template-based generation of synthetic infographics: (a) design templates; (b) synthetic infographics generated from the design templates.

## D  DATASET STATISTICS

Fig. 7 shows the distribution of the number of annotated texts, charts, and HROs per real and synthetic infographic. On average, each real infographic contains 53.76 text elements (each corresponding to a line of text), 1.94 charts, and 14.74 HROs, while each synthetic infographic contains 74.21 text elements, 3.20 charts, 10.16 HROs, and 61.69 sub-elements. The slight difference in annotation density between real and synthetic infographics enhances the diversity of the dataset, improving its utility for training models to handle diverse infographics.

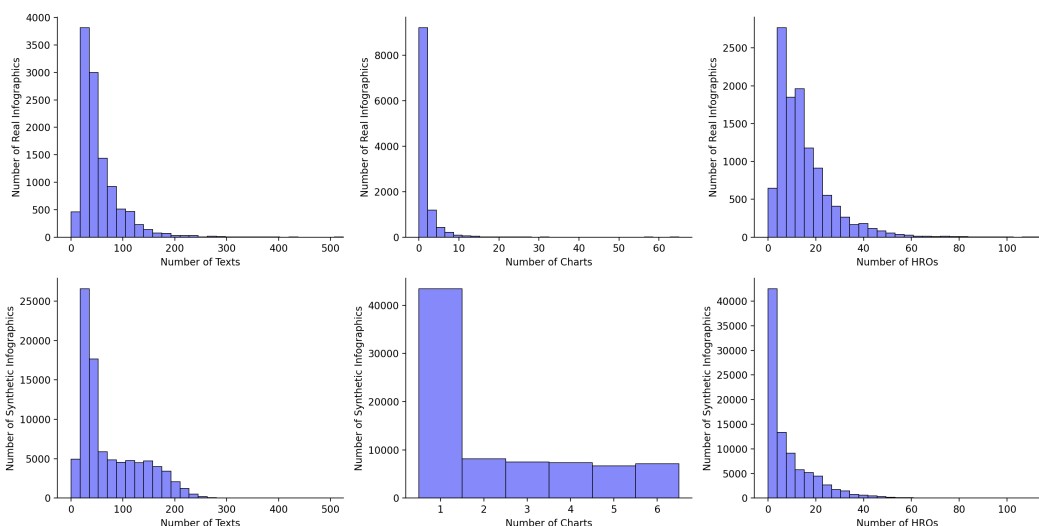

Figure 7: The distribution of the number of texts, charts, and HROs in each infographic.

We classify charts and HROs into subcategories: charts are categorized into 75 distinct types, while HROs are labeled as either data-related or theme-related objects. The 75 chart types are: 1) Vertical bar chart, 2) Vertical stacked bar chart, 3) Vertical grouped bar chart, 4) Horizontal bar chart, 5) Horizontal stacked bar chart, 6) Horizontal grouped bar chart, 7) Radial bar chart, 8) Radial stacked bar chart, 9) Radial grouped bar chart, 10) Circular bar chart, 11) Circular stacked bar chart, 12) Circular grouped bar chart, 13) Pictorial percentage bar chart, 14) Histogram, 15) Lollipop chart,

16) Dot chart, 17) Diverging bar chart, 18) Vertical bar chart with circle, 19) Horizontal bar chart with circle, 20) Vertical dot bar chart, 21) Horizontal dot bar chart, 22) Dumbbell plot, 23) Span chart, 24) Bump chart, 25) Line graph, 26) Spline graph, 27) Stepped line graph, 28) Slope chart, 29) Small multiples of line graphs, 30) Small multiples of spline graphs, 31) Small multiples of stepped line graphs, 32) Area chart, 33) Spline area chart, 34) Layered area chart, 35) Layered spline area chart, 36) Range area chart, 37) Stacked area chart, 38) Radial area chart, 39) Radial spline area chart, 40) Radial layered area chart, 41) Radial layered spline area chart, 42) Radial range area chart, 43) Radial stacked area chart, 44) Diverging area chart, 45) Diverging spline area chart, 46) Small multiples of area charts, 47) Small multiples of spline area charts, 48) Pie chart, 49) Donut chart, 50) Semicircle pie chart, 51) Semicircle donut chart, 52) Rose chart, 53) Small multiples of pie charts, 54) Small multiples of donut charts, 55) Small multiples of semicircle pie charts, 56) Small multiples of semicircle donut charts, 57) Small multiples of rose charts, 58) Radar line chart, 59) Radar spline chart, 60) Small multiples of radar line charts, 61) Small multiples of radar spline charts, 62) Proportional area chart, 63) Scatterplot, 64) Grouped scatterplot, 65) Bubble chart, 66) Heatmap, 67) Waffle chart, 68) Small multiples of waffle charts, 69) Alluvial diagram, 70) Gauge chart, 71) Small multiples of gauge charts, 72) Funnel chart, 73) Pyramid chart, 74) Treemap, 75) Voronoi treemap.

For the real infographics, we have attempted to classify the charts and HROs using GPT-4o. However, it achieves limited accuracy, with $61.49\%$ on $1,179$ charts and $74.69\%$ on $1,498$ HROs. To complement the dataset with chart-type annotations, we have attempted two more methods: 1) GPT-5; and 2) a two-stage CLIP-based linear probing method that first predicts one of 10 coarse categories (e.g., bar, line) and then refines it into one of 75 chart types using classifiers trained on our synthetic infographics. GPT-5 achieves an accuracy of $69.64\%$, while the two-stage method achieves $78.80\%$. We manually verified the chart-type annotations produced by the two-stage method and released the verified annotations on Hugging Face.

## E EXPERT BACKGROUND AND REFINEMENT ROUNDS

**Expert background**   Seven experts participated in the annotation process. Two are Ph. D. students majoring in data visualization who are co-authors of this paper. The other five are experienced annotators from a vendor company, each with at least two years of experience in object detection annotation, practical experience in plain chart annotation, and at least a bachelor's degree.

**Refinement rounds**   The annotation refinement process spanned four months, with one round lasting two weeks, resulting in a total of eight rounds. Each round followed this procedure: 1) The Ph.D. experts reviewed existing annotations to identify potential issues; 2) They trained the remaining annotators to perform targeted corrections; 3) The detection model was fine-tuned on the corrected annotations to improve subsequent predictions. We qualitatively observed that the annotation quality, as well as the detection accuracy, steadily improved across the refinement rounds. Since evaluating annotation precision and recall requires additional human effort, we evaluated precision and recall at three stages: before refinement, after the fourth round, and after all rounds. As shown in Table 8, the precision and recall steadily improved. The refinement rounds required around $3,000 in human annotation cost and 313 GPU-hours for model fine-tuning on an NVIDIA Tesla V100, priced at approximately $0.7 per GPU-hour. In total, the refinement process cost roughly $3,220. In contrast, fully manual annotation of all real infographics is estimated to cost more than $30,000, due to the expertise required and the need for multi-round quality verification to ensure annotation accuracy. It also produces a model that can generate reliable annotations for future chart-related datasets, thereby reducing annotation costs.

Table 8: Annotation precision and recall across the refinement process.

| Refinement Round | Precision | Recall |
|---|---|---|
| Before the refinement | 69.6% | 77.0% |
| After the fourth refinement round | 89.0% | 91.3% |
| After all refinement rounds | 93.9% | 96.7% |

To ensure high **inter-annotator agreement** in the refinement rounds, the five annotators were trained each round by the two Ph.D. experts, and their corrections were required to reach an average IoU of 0.95 before proceeding to annotation. To further verify the consistency of the Ph.D. annotators, we conducted an additional inter-annotator agreement study on $1,000$ real infographics. For each infographic, we first generated annotations using the model before refinement, then randomly selected one element from those annotations. The Ph.D. experts independently corrected the element, adjusting its location and category as needed. They were also asked to mark the occurrences of ambiguous chart–HRO boundaries. Table 9 shows the inter-annotator agreement between the two Ph.D. experts on all elements and on 39 elements with ambiguous chart-HRO boundaries. The experts performed consistent corrections (average IoU=0.95, category agreement=0.99). These corrections differed from the model predictions, which showed lower agreement (average IoU=0.90, category agreement=0.96), indicating that they can effectively mitigate systematic biases in the model annotations. Even for the elements with ambiguous boundaries, the agreement remains high (average IoU=0.84, category agreement=0.92), further confirming the reliability of the corrections.

| | Average IoU | Category Agreement |
|---|---|---|
| Expert-Expert | **0.95** | **0.99** |
| Expert-Model | 0.90 | 0.96 |

(a) Results on all elements

| | Average IoU | Category Agreement |
|---|---|---|
| Expert-Expert | **0.84** | **0.92** |
| Expert-Model | 0.77 | 0.91 |

(b) Results on elements with ambiguous boundaries

Table 9: Inter-annotator agreement study results between two Ph.D. experts.

## F  LIMITATIONS

**Dataset scope**   Regarding the images, InfoDet focuses on infographics that combine texts, charts, and HROs, while images lacking charts or HROs are excluded during collection. Consequently, InfoDet is less suitable as the sole training source for applications targeting graphic designs without charts or scientific documents without HROs. Regarding the annotations, a model-in-the-loop method is utilized to generate annotations for the real infographics. This method helps maintain consistent annotation guidelines across the dataset, but it may also introduce consistent errors in ambiguous cases. This should be taken into account for tasks requiring highly precise detection results.

**Remaining model errors**   At the end of the refinement process, we randomly sample $1,250$ infographics to evaluate the quality of the generated annotations. While the precision (93.9%) and recall (96.7%) are comparable to those of widely used object detection datasets, a small number of annotation errors still remain. We have summarized these errors in Table 10. The most frequent errors include: 1) imprecise bounding-box localization due to ambiguous element boundaries; 2) marks or annotations mistakenly detected as HROs due to their high visual similarity; and 3) missed tiny elements due to their limited scale.

Table 10: Distribution of remaining annotation errors after the refinement process.

| Error Type | Percentage |
|---|---|
| Imprecise bounding-box localization | 36.5% |
| Marks or annotations detected as HROs | 32.9% |
| Missed tiny elements | 20.4% |
| Others | 10.2% |

## G  CREATING MARK-LEVEL ANNOTATIONS

To create mark-level annotations, we extend our synthetic infographic creation and annotation methods. Using these methods, we generate annotations of 26 element categories. The 26 categories are: "vertical_gridline", "dumbbell_mark", "scatter_mark", "legend", "bar_mark", "proportional-area_mark", "axis", "other_gridline", "line_mark", "area-under-line_mark", "gauge_mark", "bump_mark", "horizontal_gridline", "stacked-bar_mark", "radar_mark", "donut_mark", "sankey_mark", "pie_mark", "span_mark", "bubble_mark", "histogram_mark", "treemap_mark", "waffle_mark", "pyramid_mark", "funnel_mark", and "range_mark".

## H  DATASET ANALYSIS

We have conducted analyses to verify that our dataset exhibits high diversity, no harmful bias, and high fidelity of synthetic infographics.

### H.1  DIVERSITY ANALYSIS

To evaluate the diversity of the dataset, we extract a set of attributes from each infographic and examine the variety of values for each attribute. Specifically, we consider four key attributes: chart type, infographic topic, visual style, and communication tone. Attribute values are extracted using Gemini-2.5-flash, with slightly different strategies for each attribute: 1) Chart type is selected from the 75 identified chart categories; 2) Infographic topic was categorized based on the IPTC Media Topics taxonomy, a widely adopted taxonomy for news content; 3) For visual style and communication tone, the model first generate a descriptive term for each infographic, and then grouped semantically similar terms into attribute values. We report the number of unique values identified for each attribute in Table 11. These numbers indicate a broad range of chart types, infographic topics, visual styles, and communication tones. In particular, the infographic topics span all 17 top-level categories and 96 out of 120 second-level topics in the IPTC Media Topics taxonomy. These results verify the diversity of our dataset.

Table 11: Diversity of infographic attribute values in InfoDet.

| Attribute | # Attribute Values | Example Attribute Values |
|---|---|---|
| Chart Type | 75 | horizontal bar chart, line graph, treemap |
| Infographic Topic | 96 | public health, weather forecast, market and exchange |
| Visual Style | 15 | minimalistic, cartoonish, vintage |
| Communication Tone | 21 | neutral, critical, persuasive |

### H.2  BIAS ANALYSIS

To mitigate potential bias, we have monitored attribute distributions during the infographic collection and generation and applied resampling techniques when necessary. For example, we observed an overrepresentation of bar charts in early synthetic samples and reduced their frequency through targeted resampling. We also computed pointwise mutual information (PMI) (Borenstein et al., 2023) across attribute pairs to identify unexpected co-occurrences. We have manually inspected all pairs with PMIs greater than 1 and found that the high-PMI pairs (*e.g.*, environment topic + concerned tone) aligned with common communication patterns and did not suggest harmful or misleading bias.

In summary, we found no harmful or systemic bias in the dataset and ensured that attribute variations remain reflective of real-world usage.

### H.3   QUALITY EVALUATION OF SYNTHETIC INFOGRAPHICS AND COMPARISON WITH REAL INFOGRAPHICS

**Quality evaluation**    The quality of synthetic infographics is evaluated at both the sample and dataset levels. At the sample level, we randomly select 500 synthetic infographics and manually verify their quality. All of them clearly convey the underlying data, contain HROs that align well with their intended semantics, and have accurate annotations. A minor issue is observed in 18 samples, where slight unintended overlap occurs between elements due to minor misalignments during rendering. However, this does not affect object detection, as the boundaries of the overlapping elements remain clear. At the dataset level, we evaluate how well the synthetic infographics cover the real ones in feature space. Specifically, we use CLIP (Radford et al., 2021) to extract the feature embeddings of all infographics and project them into a two-dimensional space using UMAP (McInnes et al., 2018). The space is divided into a uniform grid, and a real infographic is considered "covered" if at least one synthetic infographic falls in the same grid cell. The results show that 92.64% of the real infographics are covered, indicating the high representativeness of the synthetic infographics.

**Comparison with real infographics**    We qualitatively examine the difference in the distribution of synthetic and real infographics in the UMAP visualization (Fig. 8). While the overall distribution of synthetic and real infographics largely overlaps, some distinctive characteristics are observed. Synthetic infographics include chart variations specially crafted by design experts, such as those with hand-drawn-style fills, which are relatively rare in real infographics. Real infographics, on the other hand, uniquely feature composite charts, where multiple single charts are combined by sharing axes or overlaying each other.

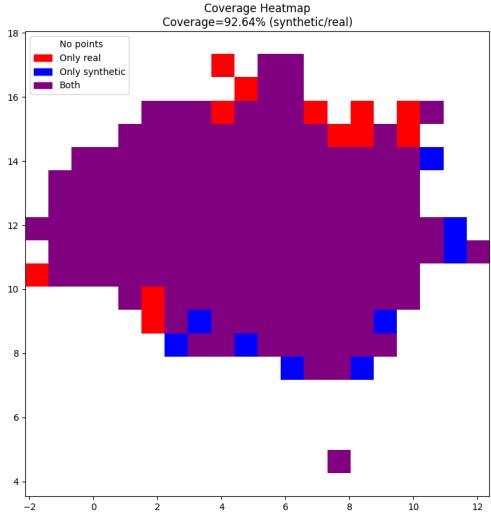

Figure 8: The distribution of synthetic and real infographics.

Additionally, we have compared the distribution of real and synthetic infographics across the attributes using Jensen-Shannon divergence (Lin, 2002), a commonly used measure of similarity between probability distributions. Table 12 shows the evaluation results. The divergences for infographic topics, visual styles, and communication tones are low, suggesting strong alignment. The divergence for chart types is relatively higher because the synthetic infographics cover all chart types and include a higher proportion of rare ones (e.g., circular bar charts, waffle charts) to ensure sufficient coverage.

Table 12: Computational costs (GPU hours) for fine-tuning models on different datasets.

| Attribute | JS Distance |
|---|---|
| Chart Type | 0.149 |
| Infographic Topic | 0.053 |
| Visual Style | 0.022 |
| Communication Tone | 0.038 |

# I DETAILED EXPERIMENTAL SETUP

## I.1 THINKING-WITH-BOXES VIA GROUNDED CHAIN-OF-THOUGHT

**Prompts for the grounded chain-of-thought method and the baselines** In the grounded chain-of-thought method, we prepend the grounded infographic elements to the question-category-specific prompt used in ChartQAPro (Masry et al., 2025). Below is an example input to the vision-language model.

---

**Example Prompt for Grounded Chain-of-Thought**

You will be provided with two versions of the same infographic chart, each with certain elements highlighted.
You will also be provided with the information lists of elements highlighted in the images. Each entry in the lists of elements follows the format (ID, Content), where:
    ID means the id of the element.
    Content means the content of the element.

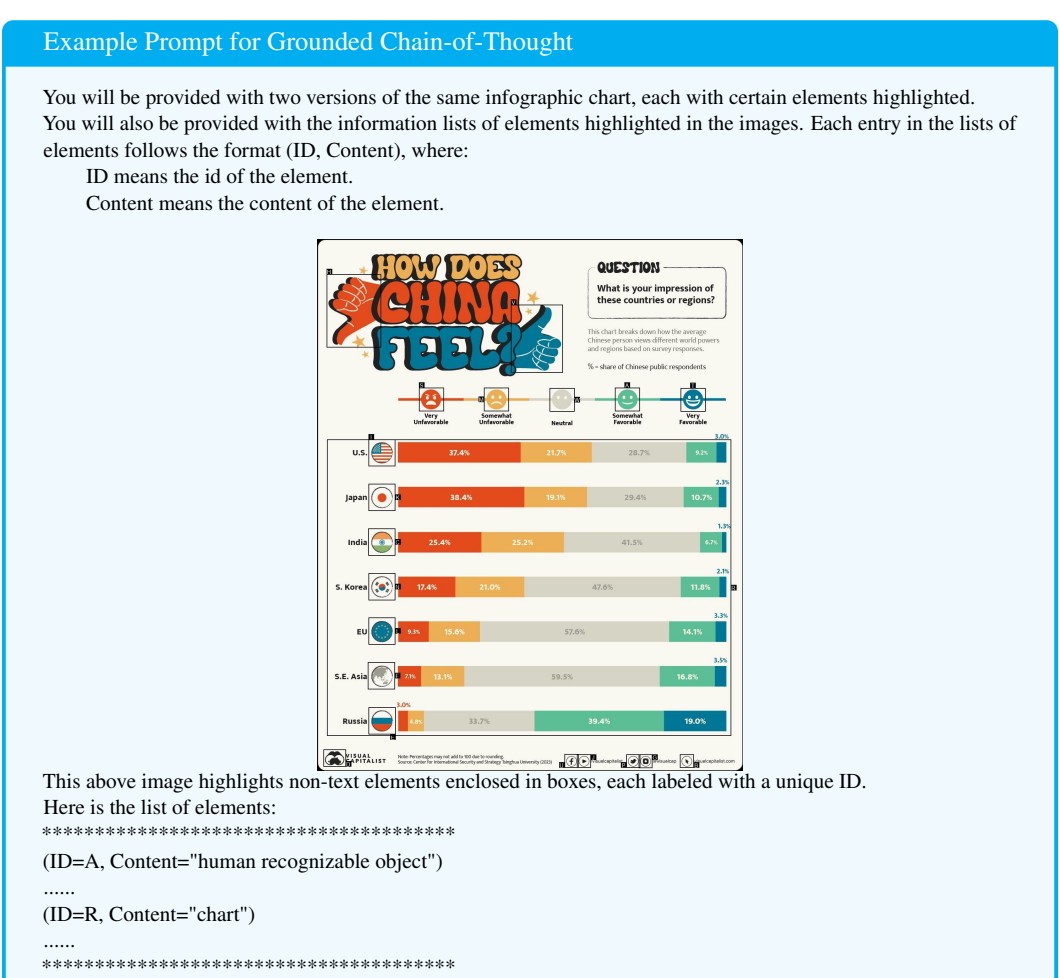

This above image highlights non-text elements enclosed in boxes, each labeled with a unique ID.
Here is the list of elements:
*************************************
(ID=A, Content="human recognizable object")
......
(ID=R, Content="chart")
......
*************************************

---

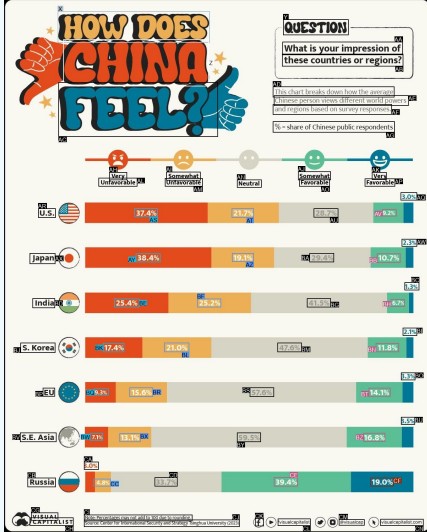

This above image highlights text elements enclosed in boxes, each labeled with a unique ID.

Here is the list of elements:

\*\*\*\*\*\*\*\*\*\*\*\*\*\*\*\*\*\*\*\*\*\*\*\*\*\*\*\*\*\*\*\*\*\*\*\*\*\*\*\*

(ID=X, Content="text: HOW DOES")

(ID=Y, Content="text: QUESTION")

......

\*\*\*\*\*\*\*\*\*\*\*\*\*\*\*\*\*\*\*\*\*\*\*\*\*\*\*\*\*\*\*\*\*\*\*\*\*\*\*\*

These labeled elements are intended to support you in your upcoming task. Please refer to and make use of them as needed during your thinking and analysis, and be sure to mention their IDs when doing so.

For example:

1. Based on the content in box ID 1, (your finding about the box), or;

2. Based on the relationships of box ID 1, ID 2, ..., ID N, (your finding based on the boxes).

Below is the image of original infographic chart, followed by your task:

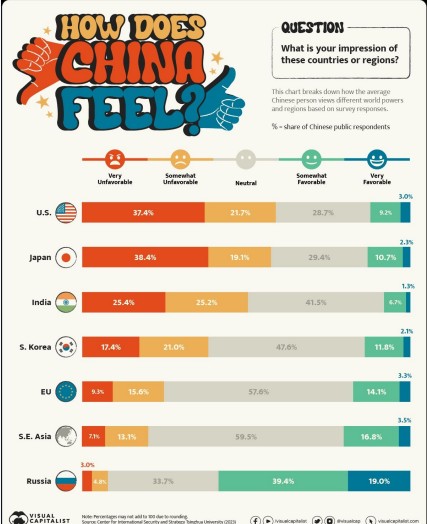

You are given a factoid question that you need to answer based on the provided image.

You need to think step-by-step, but your final answer should be a single word, number, or phrase. If the question is unanswerable based on the information in the provided image, your answer should be unanswerable. Do not generate units. But if numerical units such as million, m, billion, B, or K are required, use the exact notation shown in the chart.

If there are multiple final answers, put them in brackets using this format ['Answer1', 'Answer2'].

Remember to think step-by-step and mention the IDs of the elements you used, and reply in the following JSON format:

```
{
     "Steps": "The step-by-step thinking process with IDs mentioned.",
     "A": "Your answer."
}
```
**Question:**   What proportion of Chinese public respondents have a neutral impression of Japan?

For the baselines, we use the same prompt as ChartQAPro. Below are examples of the input for the three baselines: direct prompting, chain-of-thought, and program-of-thought.

**Example Prompt for Direct Prompting**

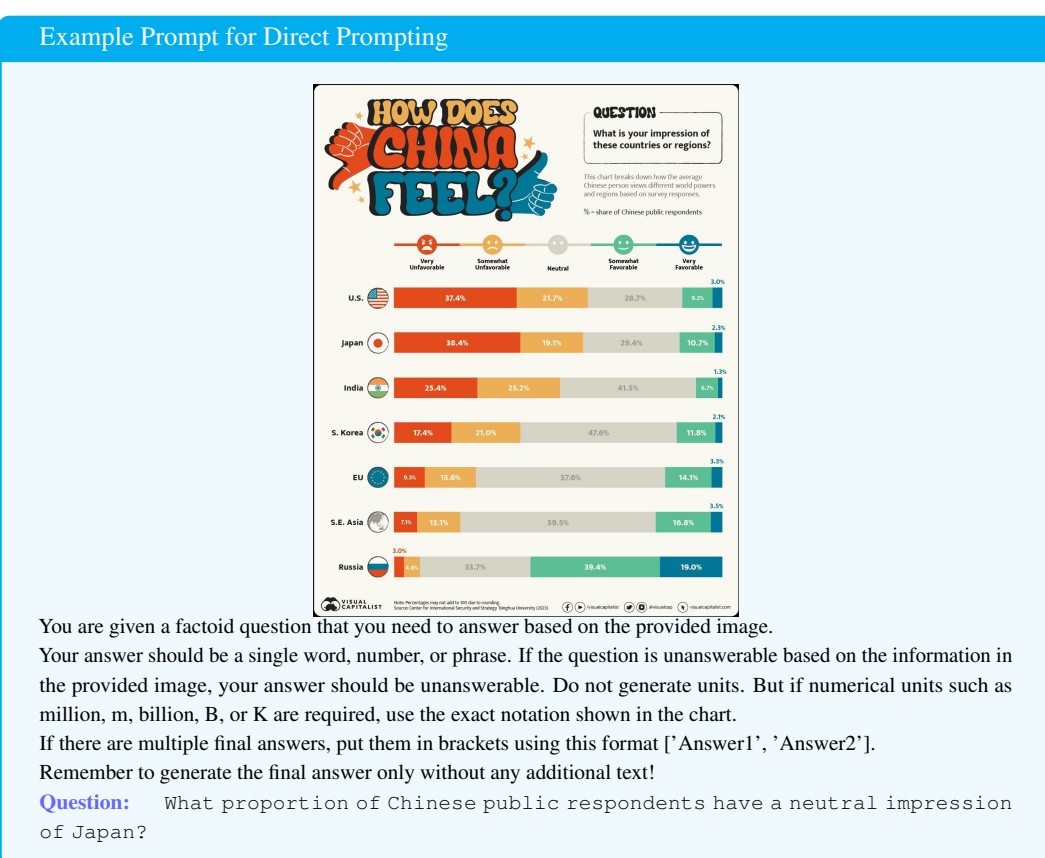

You are given a factoid question that you need to answer based on the provided image.

Your answer should be a single word, number, or phrase. If the question is unanswerable based on the information in the provided image, your answer should be unanswerable. Do not generate units. But if numerical units such as million, m, billion, B, or K are required, use the exact notation shown in the chart.

If there are multiple final answers, put them in brackets using this format ['Answer1', 'Answer2'].

Remember to generate the final answer only without any additional text!

**Question:**   What proportion of Chinese public respondents have a neutral impression of Japan?

**Example Prompt for Chain-of-Thought**

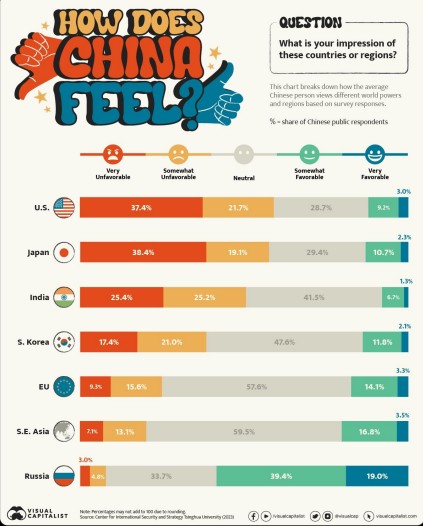

You are given a factoid question that you need to answer based on the provided image.

You need to think step-by-step, but your final answer should be a single word, number, or phrase. If the question is unanswerable based on the information in the provided image, your answer should be unanswerable. Do not generate units. But if numerical units such as million, m, billion, B, or K are required, use the exact notation shown in the chart.

If there are multiple final answers, put them in brackets using this format ['Answer1', 'Answer2'].

Remember to think step-by-step and format the final answer in a separate sentence like "The answer is X"

**Question:**  What proportion of Chinese public respondents have a neutral impression of Japan?

**Example Prompt for Program-of-Thought**

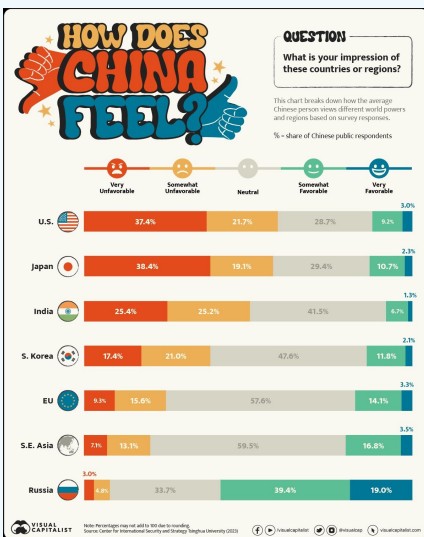

You are given a factoid question that you need to answer based on the provided image.

You need to write an executable python code that calculates and prints the final answer, but your final answer should be a single word, number, or phrase. If the question is unanswerable based on the information in the provided image, your answer should be unanswerable. Do not generate units. But if numerical units such as million, m, billion, B, or K are required, use the exact notation shown in the chart.

If there are multiple final answers, put them in brackets using this format ['Answer1', 'Answer2'].

Remember to return a python code only without any additional text.

> **Question:** What proportion of Chinese public respondents have a neutral impression of Japan?

**Comparison of the visual prompts rendered in one layer versus two layers**  In our grounded chain-of-thought method, we propose to separate the visual prompts into two layers: one for charts and HROs, and the other for texts. As shown in Fig. 9, this separation improves visual clarity by reducing overlap between bounding boxes.

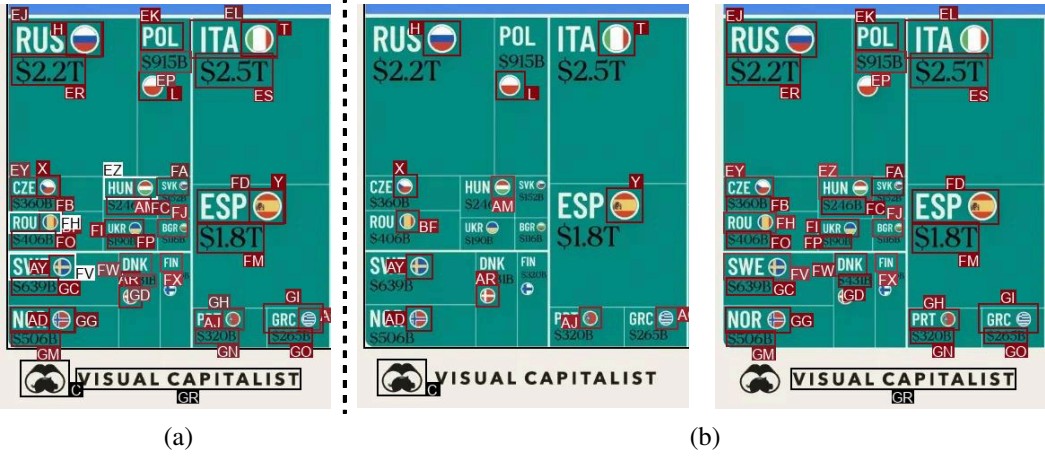

(a)                                        (b)

Figure 9: Comparison of the visual prompts rendered with different layer configurations: (a) visual prompts rendered in one merged layer: (b) visual prompts rendered in two separate layers.

**Enhanced relaxed accuracy implementation**  Following ChartQAPro, we use the enhanced relaxed accuracy to evaluate the chart understanding performance. This metric evaluates answers based on the following criteria:

1. Numeric answers are allowed a 5% error margin.
2. For answers in 'years', an exact match is required.
3. Textual answers are evaluated using the ANLS score (Biten et al., 2019), which is based on the edit distance between texts.
4. Multiple-choice and fact-checking tasks are evaluated using an exact-match criterion.

To more accurately evaluate model performance, we make three refinements to the official implementation of the enhanced relaxed accuracy:

1. We remove punctuation marks (*i.e.*, commas and periods) from answers, ensuring that '25,000' and '25000' are treated as equivalent.
2. We remove unit symbols when evaluating numeric answers, so that values like '100' and '$100' are treated as equivalent.
3. We standardize ratios and percentages by converting them into decimal form, so that expressions like '3:2', '150%', and '1.5' are all treated as equivalent.

## I.2 COMPARING OBJECT DETECTION MODELS

We evaluate the performance of existing object detection models in detecting charts and HROs. As the models are not tailored to detecting charts and HROs, we adapt them using three adaptation methods: 1) **Zero-shot prompting**, which uses text prompts to define target classes, 2) **Few-shot prompting**, which uses $k$ randomly selected infographics to describe target classes, optionally augmented with text prompts, and 3) **Standard fine-tuning**, which updates model weights using annotated infographics, either with $k$ random example infographics or the InfoDet training set.

For zero-shot prompting, we evaluate six models: RegionCLIP (Zhong et al., 2022), Detic (Zhou et al., 2022), Grounding DINO (Liu et al., 2024), GLIP (Li et al., 2022b), MQ-GLIP (Xu et al., 2023),

and DINO-X (Ren et al., 2024), all of which take the class names "chart" and "human recognizable object" as input.

For few-shot prompting, we evaluate two models: T-Rex2 (Jiang et al., 2024) and MQ-GLIP (Xu et al., 2023). For T-Rex2, we provide $k$ randomly selected infographics with bounding box annotations. For MQ-GLIP, we provide the class names along with the selected infographics.

For traditional fine-tuning, we evaluate six models: RegionCLIP, Detic, Faster R-CNN (Ren et al., 2015), YOLOv3 (Redmon & Farhadi, 2018), RTMDet (Lyu et al., 2022), and Co-DETR (Zong et al., 2023). For fine-tuning on the entire InfoDet training set, we train for $E$ epochs with a batch size of $\mathcal{B}$ and a learning rate of $lr$. Table 13 shows the fine-tuning hyperparameters, which adhere to the official settings, as well as the computational costs, in terms of GPU hours using NVIDIA GeForce RTX 4090 D. For few-shot fine-tuning, we adjust the number of training epochs inversely with the number of random infographics, ensuring consistent computational costs. All other fine-tuning hyperparameters remain unchanged.

Table 13: Training hyperparameters and computational costs for traditional fine-tuning on the entire InfoDet training set.

| Hyperparameters | RegionCLIP | Detic | Faster R-CNN | YOLOv3 | RTMDet | Co-DETR |
|---|---|---|---|---|---|---|
| Optimizer | SGD | AdamW | SGD | SGD | AdamW | AdamW |
| $E$ | 1 | 8 | 10 | 10 | 5 | 3 |
| $\mathcal{B}$ | 1 | 8 | 64 | 64 | 64 | 64 |
| $lr$ | $5e-4$ | $3.75e-6$ | $2e-3$ | $1e-3$ | $4e-3$ | $1e-5$ |
| Computational costs (GPU hours) | 20 | 40 | 20 | 30 | 40 | 70 |

We have compared the computational costs of fine-tuning the detection models on InfoDet and existing standard datasets, including PASCAL VOC (Everingham et al., 2010), COCO (Lin et al., 2014), LVIS (Gupta et al., 2019), and Objects365 (Shao et al., 2019) in Table 14. The computational cost of fine-tuning on InfoDet is higher than that of Pascal VOC, comparable to that of COCO, and lower than that of Objects365 and LVIS.

Table 14: Computational costs (GPU hours) for fine-tuning models on different datasets.

| Dataset | RegionCLIP | Detic | Faster R-CNN | YOLOv3 | RTMDet | Co-DETR |
|---|---|---|---|---|---|---|
| PASCAL VOC | 3 | 6 | 3 | 5 | 6 | 11 |
| COCO | 23 | 47 | 23 | 35 | 47 | 82 |
| LVIS | 32 | 65 | 32 | 49 | 65 | 113 |
| Objects365 | 119 | 238 | 119 | 179 | 238 | 416 |
| InfoDet | 20 | 40 | 20 | 30 | 40 | 70 |

### I.3 APPLYING THE DEVELOPED MODEL TO GRAPHIC LAYOUT DETECTION

We evaluate the InternImage-based model on three graphic layout detection datasets, Rico (Deka et al., 2017) and DocGenome (Xia et al., 2024), and PosterLayout (Hsu et al., 2023). Rico contains over 66K user interfaces collected from Android applications. Following the common practice (Manandhar et al., 2020; 2021), we aim to detect 25 UI component classes and split the dataset into 53K layouts for training and 13K for testing. DocGenome is a large-scale scientific document dataset of 6.8M pages sourced from the arXiv repository, annotated with bounding boxes for 13 categories of components. We randomly sample 113K pages for training and 13K for testing. PosterLayout contains 9,974 posters annotated with bounding boxes of text, logo, and underly. We randomly split the dataset into 8,974 for training and 1,000 for testing. We fine-tune five model variants for each dataset. Four of which are pre-trained on a different combination of ImageNet-22K (Deng et al., 2009), Objects365 (Shao et al., 2019), COCO (Lin et al., 2014), and InfoDet. The fifth variant is pre-trained on all four datasets and leverages a hierarchy-guided matching (HGM) scheme on InfoDet that

encourages matched predictions to preserve the hierarchical relationships between charts and their sub-elements in the ground truth. Specifically, in the DINO detector, Hungarian matching determines one-to-one correspondences between predictions and ground-truth boxes for training loss computation. Prior works such as DN-DETR (Li et al., 2022a) and Stable-DINO (Liu et al., 2023) have shown that a reliable matching process is crucial for stable model optimization. Motivated by this, we incorporate hierarchy constraints, where chart sub-elements should be spatially contained within charts, into the matching process to make it more reliable. However, directly incorporating such hierarchy constraints turns the matching into a quadratic assignment problem, which is NP-hard and intractable to solve in practice. Therefore, we adopt a two-stage matching method, first performing standard Hungarian matching and then refining the matching iteratively to satisfy the hierarchy constraints. Following the official setting (Wang et al., 2023), we fine-tune the frozen InternImage backbones along with the DINO detector (Zhang et al., 2023) for 12 epochs. The batch size is set to 16, and we use an AdamW optimizer (Loshchilov & Hutter, 2019) with an initial learning rate of 0.0001 and a weight decay of 0.05. We use a step-based learning rate scheduler, which decreases the learning rate by a factor of 0.1 at epochs 8 and 11. The training takes 196 GPU hours on Rico, 296 GPU hours on DocGenome, and 12 GPU hours on PosterLayout using NVIDIA Tesla V100.

## J    ADDITIONAL RESULTS AND ANALYSIS OF THINKING-WITH-BOXES

### J.1    RESULTS ON CHARTQA AND PLOTQA

To test the generalizability of our method beyond ChartQAPro, we compare it against direct prompting, the best-performing baseline, on ChartQA (Hoque et al., 2022) and PlotQA (Methani et al., 2020) using o1. Table 15 shows the evaluation results. Our method performs better than direct prompting on both ChartQA and PlotQA, indicating that its benefits extend beyond ChartQAPro.

| Method | ChartQA | PlotQA |
|---|---|---|
| Direct | 75.6 | 30.6 |
| Grounded CoT (ours) | **79.2** | **31.3** |

Table 15: Performance of o1 on ChartQA and PlotQA.

### J.2    ADDITIONAL STATISTICAL ANALYSIS

To assess the robustness of the improvement, we conducted a repeated set of runs on o3, the model with the smallest improvement. As shown in Table 16, in both runs, our method achieves the highest accuracy among all prompting methods. Even when accounting for variance, its mean performance remains clearly above the baselines.

| | Direct | CoT | PoT | Grounded CoT (ours) |
|---|---|---|---|---|
| The first run | 60.6 | 60.0 | 59.5 | **61.6** |
| The second run | 61.4 | 60.1 | 61.0 | **62.3** |
| Mean ± Std | $61.0 \pm 0.5$ | $60.0 \pm 0.1$ | $60.3 \pm 1.0$ | **$62.0 \pm 0.5$** |

Table 16: Performance of o3 across two runs. The best score in each row is **bold**.

### J.3 ADDITIONAL QUALITATIVE ANALYSIS

Fig. 10 shows how the provided grounded annotations elicit the model's reasoning capabilities. In Fig. 10 (a), direct prompting produces the incorrect answer "Switzerland" because it simply reads the second label. In contrast, grounded Chain-of-Thought first identifies the second-shortest bar and then assigns it to the correct label "Japan". Similarly, in Fig. 10 (b), direct prompting misinterprets the relative bar heights between 2008 and 2013, while grounded Chain-of-Thought correctly compares the visual differences by performing multi-step reasoning over them. These examples suggest that grounded Chain-of-Thought leverages the model's visual reasoning capabilities, mitigating errors that arise from relying on superficial visual cues rather than depending solely on prompt design.

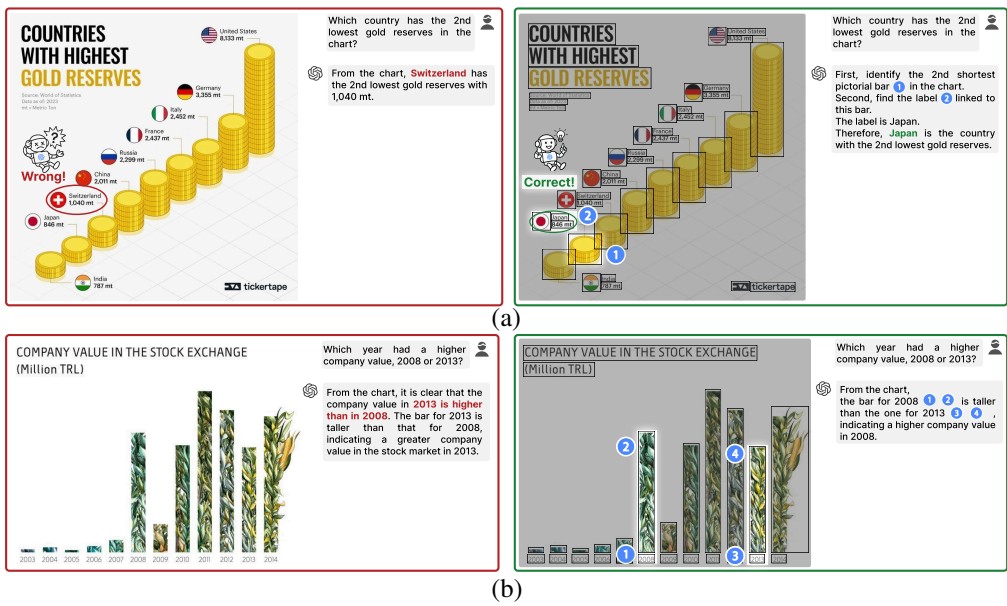

Figure 10: Additional qualitative comparison between direct prompting and grounded CoT.

### J.4 DETAILED ANALYSIS OF ERRORS BY O3 ON CHARTQAPRO

Despite its strong visual reasoning capability, o3 achieves slightly lower accuracy compared to o1 and o4-mini on the ChartQAPro benchmark (Masry et al., 2025). To investigate this, we randomly sample 200 question-answer pairs and analyze the failure patterns when using grounded CoT. We identify two primary sources of failures: 1) **perception error**, where models fail to correctly interpret the content and relationships of the infographic elements, and 2) **instruction following error**, where models do not adhere to the prompt when formatting the answer. As shown in Table 17, perception errors are the main cause of chart understanding failures, occurring with similar frequency across all models. However, o3 shows a higher frequency of instruction-following errors, contributing to its slightly lower overall performance compared to o1 and o4-mini. In particular, even when instructed to output the numerical answer as a single word, o3 often includes extra words like '≈' and 'about'. To address this, we have attempted to increase the reasoning effort from 'medium' to 'high'. However, as shown in Table 18, this change does not yield an obvious improvement in the chart understanding performance, and the instruction following error still occurs with a similar frequency. This suggests that the 'medium' setting already provides sufficient reasoning budget for ChartQAPro, and alternative strategies are needed to enhance o3's instruction-following ability.

Table 17: Error analysis of chart understanding failures on ChartQAPro for o1, o3, and o4-mini.

| Model | Perception Error | Instruction Following Error |
|---|---|---|
| o1 | 48 | 12 |
| o3 | 47 | 22 |
| o4-mini | 46 | 8 |

Table 18: Performance of o3 using different levels of reasoning effort.

| Reasoning Effort | Direct | CoT | PoT | Grounded CoT |
|---|---|---|---|---|
| Medium | 60.6 | 60.0 | 59.5 | 61.6 |
| High | 60.4 | 61.0 | 60.8 | 61.8 |

### J.5 ADDITIONAL ABLATION STUDIES

**Varying detection model**   We replace our InternImage-based detector with two alternatives: Faster R-CNN, which performs worse on COCO, and Co-DETR, which performs comparably on COCO. As shown in Table 19(a), the accuracy on ChartQAPro drops when using Faster R-CNN, while it remains comparable when using Co-DETR, indicating that Grounded CoT benefits from stronger detectors and remains stable when the underlying detection quality is high.

**Varying OCR model**   We replace our PP-OCRv4 with two alternatives: EasyOCR (JaidedAI, 2024), a commonly used baseline OCR model, and AzureOCR (Microsoft, 2024), one of the strongest commercial OCR systems. As shown in Table 19(b), the accuracy on ChartQAPro drops when using EasyOCR, while it remains comparable when using AzureOCR, indicating that Grounded CoT can benefit from stronger OCR to some extent, but PP-OCRv4 is already sufficiently strong for this task.

| Detection model | ChartQAPro |
|---|---|
| Faster R-CNN (weaker) | 63.4 |
| Co-DETR (comparable) | 63.8 |
| Ours | **64.1** |

(a) Detection model

| OCR model | ChartQAPro |
|---|---|
| EasyOCR (weaker) | 62.9 |
| Ours | **64.1** |
| AzureOCR (stronger) | 64.0 |

(b) OCR model

Table 19: Additional ablation studies on o1.

## K  DETAILED EVALUATION RESULTS AND ANALYSIS

### K.1  DETAILED EVLUATION RESULTS

**Comparing adaptation methods and object detection models**   We evaluate all applicable adaptation methods for each model, except for standard fine-tuning, which is restricted to models that fit within the memory constraints of an Nvidia Tesla V100 GPU. For few-shot prompting and fine-tuning methods, we use $k = 4$, 10, and 30 randomly selected infographics. We average the results over 3 runs, excluding T-Rex2 and DINO-X, due to their reliance on charged APIs. Tables 20 and 21 show the AP and AR along with their standard deviation for all models.

Table 20: AP of object detection models for the chart and HRO categories. The best one is **bold**.

| | Model | Zero-shot prompting | Few-shot prompting | | | Standard fine-tuning | | | |
|---|---|---|---|---|---|---|---|---|---|
| | | | 4-shots | 10-shots | 30-shots | 4-shots | 10-shots | 30-shots | InfoDet |
| **Chart Category** | | | | | | | | | |
| Foundation Models | RegionCLIP | 0.83 | - | - | - | 6.81 ± 3.70 | 9.38 ± 1.62 | 10.12 ± 0.96 | 10.14 ± 0.80 |
| | Detic | 1.77 | - | - | - | 19.62 ± 5.61 | 23.02 ± 1.38 | 28.00 ± 1.42 | 39.57 ± 0.10 |
| | Grounding Dino | 12.64 | - | - | - | - | - | - | - |
| | GLIP | 13.52 | - | - | - | - | - | - | - |
| | MQ-GLIP | 13.52 | 16.24 ± 0.63 | 16.80 ± 0.19 | 16.91 ± 0.24 | - | - | - | - |
| | T-Rex2 | - | 12.22 | - | - | - | - | - | - |
| | DINO-X | **13.99** | - | - | - | - | - | - | - |
| Traditional Models | Faster R-CNN | - | - | - | - | 3.43 ± 2.46 | 5.63 ± 2.34 | 9.89 ± 2.89 | 78.92 ± 0.33 |
| | YOLOv3 | - | - | - | - | 5.49 ± 1.64 | 5.12 ± 2.81 | 7.83 ± 2.04 | 14.70 ± 5.92 |
| | RTMDet | - | - | - | - | 12.82 ± 1.74 | 25.90 ± 2.44 | 28.84 ± 2.09 | 83.65 ± 3.46 |
| | Co-DETR | - | - | - | - | **27.63 ± 11.41** | **31.65 ± 4.20** | **43.36 ± 2.99** | **88.22 ± 0.64** |
| **HRO Category** | | | | | | | | | |
| Foundation Models | RegionCLIP | 3.57 | - | - | - | 14.70 ± 0.55 | 18.28 ± 0.23 | 18.80 ± 1.38 | 23.26 ± 0.31 |
| | Detic | 4.38 | - | - | - | 14.24 ± 5.15 | 22.47 ± 3.29 | 30.41 ± 1.19 | 34.31 ± 0.59 |
| | Grounding Dino | 12.24 | - | - | - | - | - | - | - |
| | GLIP | 11.21 | - | - | - | - | - | - | - |
| | MQ-GLIP | 11.21 | 15.46 ± 1.43 | 16.18 ± 0.36 | 16.94 ± 0.29 | - | - | - | - |
| | T-Rex2 | - | 16.15 | - | - | - | - | - | - |
| | DINO-X | **14.94** | - | - | - | - | - | - | - |
| Traditional Models | Faster R-CNN | - | - | - | - | 0.98 ± 0.13 | 4.07 ± 1.05 | 11.59 ± 1.01 | 49.02 ± 0.17 |
| | YOLOv3 | - | - | - | - | 3.96 ± 0.77 | 5.95 ± 1.52 | 9.14 ± 1.28 | 25.52 ± 2.48 |
| | RTMDet | - | - | - | - | 18.91 ± 2.41 | 21.91 ± 1.34 | 26.38 ± 3.29 | 53.64 ± 0.22 |
| | Co-DETR | - | - | - | - | **25.46 ± 2.43** | **31.09 ± 1.10** | **36.94 ± 3.62** | **64.02 ± 4.73** |

Table 21: AR of object detection models for the chart and HRO categories. The best one is **bold**.

| | Model | Zero-shot prompting | Few-shot prompting | | | Standard fine-tuning | | | |
|---|---|---|---|---|---|---|---|---|---|
| | | | 4-shots | 10-shots | 30-shots | 4-shots | 10-shots | 30-shots | InfoDet |
| **Chart Category** | | | | | | | | | |
| Foundation Models | RegionCLIP | 13.93 | - | - | - | 15.50 ± 3.22 | 19.37 ± 1.47 | 19.54 ± 0.81 | 17.48 ± 0.65 |
| | Detic | 23.72 | - | - | - | 36.99 ± 6.02 | 40.40 ± 0.97 | 44.06 ± 1.39 | 57.38 ± 0.41 |
| | Grounding Dino | **63.22** | - | - | - | - | - | - | - |
| | GLIP | 44.89 | - | - | - | - | - | - | - |
| | MQ-GLIP | 44.88 | 43.47 ± 0.46 | 43.81 ± 0.51 | 43.97 ± 0.22 | - | - | - | - |
| | T-Rex2 | - | 21.84 | - | - | - | - | - | - |
| | DINO-X | 29.36 | - | - | - | - | - | - | - |
| Traditional Models | Faster R-CNN | - | - | - | - | 10.79 ± 4.69 | 15.86 ± 2.74 | 20.86 ± 4.79 | 80.84 ± 0.31 |
| | YOLOv3 | - | - | - | - | 16.21 ± 2.54 | 15.98 ± 5.08 | 23.33 ± 1.55 | 43.16 ± 5.45 |
| | RTMDet | - | - | - | - | 44.16 ± 0.67 | 53.48 ± 4.74 | 56.34 ± 1.92 | 86.41 ± 0.25 |
| | Co-DETR | - | - | - | - | **53.41 ± 12.34** | **61.10 ± 6.36** | **68.36 ± 2.80** | **89.82 ± 0.53** |
| **HRO Category** | | | | | | | | | |
| Foundation Models | RegionCLIP | 24.92 | - | - | - | 22.85 ± 1.55 | 26.21 ± 0.65 | 27.08 ± 0.43 | 28.56 ± 0.27 |
| | Detic | 11.31 | - | - | - | 22.80 ± 6.66 | 34.52 ± 5.80 | 46.16 ± 1.44 | 47.74 ± 0.50 |
| | Grounding Dino | **45.97** | - | - | - | - | - | - | - |
| | GLIP | 33.21 | - | - | - | - | - | - | - |
| | MQ-GLIP | 33.20 | 40.72 ± 2.06 | 41.69 ± 1.45 | 42.38 ± 0.54 | - | - | - | - |
| | T-Rex2 | - | 24.72 | - | - | - | - | - | - |
| | DINO-X | 29.14 | - | - | - | - | - | - | - |
| Traditional Models | Faster R-CNN | - | - | - | - | 1.55 ± 1.01 | 8.77 ± 3.56 | 25.92 ± 2.03 | 52.69 ± 0.21 |
| | YOLOv3 | - | - | - | - | 13.06 ± 1.76 | 18.75 ± 1.03 | 25.31 ± 0.98 | 35.70 ± 2.01 |
| | RTMDet | - | - | - | - | 49.10 ± 0.69 | 49.72 ± 0.94 | 52.50 ± 1.15 | 59.94 ± 0.14 |
| | Co-DETR | - | - | - | - | **49.74 ± 0.35** | **57.15 ± 0.38** | **61.48 ± 0.84** | **69.47 ± 0.92** |

**Cross-dataset evaluation between our dataset and VG-DCU** We evaluate the generalizability of the traditional models via cross-dataset evaluation following Deng et al. (2023). Specifically, we train each model on either our infographics or VG-DCU (Dou et al., 2024), which comprises plain charts with element-level annotations, and evaluate each model on the other dataset. To support the transfer between datasets, we identify common categories with identical annotation guidelines, resulting in four categories: "bar_mark", "line_mark", "pie_mark", and "axis". We train and evaluate the models on these categories. Table 22 shows the evaluation results. Models show a smaller drop in mAP when transferred from our infographics to VG-DCU compared to the opposite direction. This shows that models trained on our data exhibit stronger generalizability due to the inclusion of infographic charts.

Table 22: Cross-dataset evaluation results

| Training set | VG-DCU | | | Ours | | |
|---|---|---|---|---|---|---|
| Test set | VG-DCU | Ours | mAP↓ | Ours | VG-DCU | mAP↓ |
| Faster R-CNN | 59.5 | 19.2 | 40.3 | 53.7 | 19.1 | **34.6** |
| YOLOv3 | 26.4 | 4.1 | 22.3 | 23.2 | 12.5 | **10.7** |
| RTMDet | 70.2 | 28.1 | 42.1 | 60.9 | 42.4 | **18.5** |
| Co-DETR | 86.7 | 33.0 | 53.7 | 71.0 | 53.9 | **17.1** |

## K.2 ADDITIONAL ANALYSIS

**Post-fine-tuning regression on natural images** Our object detection model is built by fine-tuning InternImage-L with the DINO detector. After fine-tuning, the model no longer predicts natural-image classes, making direct evaluation on COCO or other natural-image benchmarks infeasible. Developing a unified model that robustly handles both infographics and natural images would indeed be highly beneficial, but it is beyond the scope of our current work, which focuses exclusively on understanding infographics.

**Mark-level performance drop** The drop in detection performance from element-level to mark-level is primarily due to two factors: 1) The mark-level annotations contain more classes with high visual similarity (e.g., "gauge_mark" and "donut_mark"), which increases classification difficulty; and 2) The mark-level annotations contain smaller chart sub-elements, which makes localization harder. Boundary ambiguity is not a critical factor because most mark-level categories correspond to clear geometric shapes with well-defined boundaries.

**Why HRO detection is harder** Analysis of the remaining errors of our InternImage-based model in Appendix F reveals that the most frequent errors include: 1) imprecise bounding-box localization due to ambiguous element boundaries (36.5%); 2) marks or annotations mistakenly detected as HROs due to their high visual similarity (32.9%); and 3) missed tiny elements due to their limited scale (20.4%). The first and third errors occur more often for HROs, while the second error occurs only for HROs, making them harder to detect.

**Which HRO type is more challenging** Data-related HROs are more challenging to detect for three reasons: 1) They are more likely to be closely integrated with charts, resulting in ambiguous boundaries; 2) Marks and annotations are more likely to be mistakenly detected as data-related HROs; and 3) They are usually smaller and harder to localize.

**Root cause of foundation model failure** While foundation models achieve limited performance (mAP < 20), traditional models fine-tuned on our dataset can achieve much higher performance (mAP > 75) despite weaker inherent capabilities. This suggests that the primary cause of foundation model failure is pre-training data mismatch, rather than the intrinsic difficulty of the task.

**Rico/DocGenome components most improved by InfoDet pre-training** Table 23 shows the AP changes of Rico and DocGenome component types when fine-tuned from scratch versus pre-trained on InfoDet. The top three component types benefiting most from InfoDet pre-training are Date Picker (+29.5), Slider (+17.4), and Radio Button (+16.6) in Rico, and Text (+11.0), Code (+8.5), and Title (+7.6) in DocGenome. We observe that the improvement on Rico (42.1→50.6, Δ=8.5) is larger than that on DocGenome (69.0→74.4, Δ=5.4). This is because Rico's data contains more complex UI

hierarchies, which align well with InfoDet's chart and sub-element hierarchies and can thus better leverage the pre-training, whereas DocGenome components are largely flat.

| Category | $\Delta$AP |
|---|---|
| Date Picker | 29.5 |
| Slider | 17.4 |
| Radio Button | 16.6 |
| Card | 13.5 |
| Number Stepper | 13.5 |
| Advertisement | 12.8 |
| Checkbox | 10.5 |
| On/Off Switch | 8.9 |
| Input | 8.7 |
| Background Image | 7.8 |
| List Item | 6.9 |
| Text | 6.8 |
| Text Button | 6.5 |
| Map View | 6.5 |
| Icon | 5.8 |
| Modal | 5.8 |
| Image | 5.7 |
| Web View | 5.2 |
| Multi-Tab | 4.7 |
| Pager Indicator | 4.7 |
| Bottom Navigation | 4.0 |
| Video | 3.6 |
| Drawer | 2.8 |
| Toolbar | 2.2 |
| Button Bar | 0.4 |

(a) Rico

| Category | $\Delta$AP |
|---|---|
| Text | 11.0 |
| Code | 8.5 |
| Title | 7.6 |
| Caption | 6.5 |
| Footnote | 6.4 |
| Equation | 5.9 |
| Text-EQ | 5.7 |
| Figure | 4.4 |
| Algorithm | 4.1 |
| List | 4.1 |
| Abstract | 2.9 |
| Table | 0.0 |
| PaperTitle | -2.5 |

(b) DocGenome

Table 23: AP changes on Rico and DocGenome component types after InfoDet pre-training.

## L  THE USE OF LARGE LANGUAGE MODELS

The use of LLMs in this work is limited to the following: 1) polishing the writing for clarity; 2) filtering candidate infographics during dataset construction.

