# OpenReview forum: "InfoDet: A Dataset for Infographic Element Detection"
_ICLR.cc/2026/Conference — ICLR 2026 Poster_

### Official Review · Reviewer_x562 · 2025-10-31

**Soundness:** 2
**Presentation:** 2
**Contribution:** 2
**Rating:** 6
**Confidence:** 4

**Summary:**

The paper introduces InfoDet, a large-scale dataset for detecting infographic elements， including charts, text, and human-recognizable objects (HROs), to improve visual grounding for chart understanding. InfoDet combines 11,264 real and 90,000 synthetic infographics with over 14 million bounding-box annotations, produced via a hybrid programmatic + model-in-the-loop pipeline. The paper demonstrates InfoDet's value via three applications: (1) enabling a "Thinking-with-Boxes" (Grounded CoT) scheme that significantly improves VLM performance on the ChartQAPro benchmark (+49.6 Enhanced Relaxed Acc. for O4-mini, Fig 1); (2) benchmarking object detection models, showing InfoDet-trained models outperform SOTA foundation models ; and (3) showing the InfoDet-trained detector generalizes to UI and document layout tasks.

**Strengths:**

1. This work introduces InfoDet, a large-scale dataset featuring 101k images and 14M annotations, specifically designed to address the complex layouts of infographics that integrate charts and Human-Related Objects (HROs).
2. The usefulness of the dataset is validated across diverse downstream tasks (Section 4). The authors provide compelling quantitative evidence for its impact on VLM grounding (+49.6 gain on ChartQAPro, Sec 4.1), its utility as a challenging benchmark for object detectors (Table 4), and the robust generalizability of its features to out-of-domain tasks (Table 5)..

**Weaknesses:**

1. The model-in-the-loop method (Sec 3.2) may introduce systematic biases from the initial model (trained on synthetic data) into the real-image annotations. Please discuss this limitation and ideally quantify it, perhaps by reporting Inter-Annotator Agreement (IAA) among experts or by comparing against a small, fully-manual gold-standard set.
2. The method layers visual overlays and textual descriptors, but it’s unclear how much gain comes from (i) two-layer overlays, (ii) textual element lists, or (iii) better detections/OCR. Please include controlled ablations that independently remove the text lists, the two-layer separation, and vary detector/OCR quality.
3. The paper does not include a dedicated limitations discussion. A concise reflection on dataset scope and failure modes would improve transparency and guide responsible use.

**Questions:**

N/A

---

> ### Author Response · Authors · 2025-11-24
> **Response to Reviewer x562 (1/2)**
>
> We sincerely appreciate your valuable and helpful comments.
> Per your suggestions, we have carefully addressed the identified weaknesses.
> Please see our response below.
>
> > **Weakness 1**: The model-in-the-loop method (Sec 3.2) may introduce systematic biases from the initial model (trained on synthetic data) into the real-image annotations. Please discuss this limitation and ideally quantify it, perhaps by reporting Inter-Annotator Agreement (IAA) among experts or by comparing against a small, fully-manual gold-standard set.
>
> **A**: We have now discussed this limitation in **Appendix F**, noting that the model-in-the-loop method may introduce systematic errors in ambiguous cases.
> To evaluate how effectively experts can correct these potential errors, we conducted an inter-annotator agreement study on 1,000 real infographics.
> For each infographic, we first generated annotations using the initial model, then randomly selected one element from those annotations.
> Two experts independently corrected the element, adjusting its location and category as needed.
> They were also asked to mark the occurrences of ambiguous chart–HRO boundaries.
> The results on all elements and on 39 elements with ambiguous chart-HRO boundaries are as follows.
> The experts performed consistent corrections (average IoU=0.95, category agreement=0.99).
> These corrections differed from the model predictions, which showed lower agreement (average IoU=0.90, category agreement=0.96), indicating that they can effectively mitigate systematic biases in the model annotations.
> Even for the elements with ambiguous boundaries, the agreement remains high (average IoU=0.84, category agreement=0.92), further confirming the reliability of the corrections.
> We have included this study in **Appendix E**.
>
> |All Elements|Average IoU|Category Agreement|
> |-|-|-|
> |Expert-Expert|**0.95**|**0.99**|
> |Expert-Model|0.90|0.96|
>
> |Elements with Ambiguous Boundaries|Average IoU|Category Agreement|
> |-|-|-|
> |Expert-Expert|**0.84**|**0.92**|
> |Expert-Model|0.77|0.91|
>
> > **Weakness 2**: The method layers visual overlays and textual descriptors, but it’s unclear how much gain comes from (i) two-layer overlays, (ii) textual element lists, or (iii) better detections/OCR. Please include controlled ablations that independently remove the text lists, the two-layer separation, and vary detector/OCR quality.
>
> **A**: We have provided the ablations that independently remove the text lists and the two-layer separation in **Sec. 4.1.3** and those that independently vary detector and OCR quality in **Appendix J.5**.
>
> **Remove text lists**:
> As shown in the following table, removing the text lists results in a performance drop from 64.1 to 62.8.
> This highlights the complementary roles of visual prompts and textual descriptions in grounding infographic elements and supporting VLMs in chart understanding.
>
> |Visual|Visual+Textual|
> |-|-|
> |62.8|**64.1**|
>
> **Remove two-layer separation**:
> As shown in the following table, combining all prompts in a single layer instead of separating them results in a performance drop from 64.1 to 62.3.
> This suggests that reducing overlap through separation facilitates the visual grounding of infographic elements and improves chart understanding.
>
> |1-Layer|2-Layer|
> |-|-|
> |62.3|**64.1**|
>
> **Vary detector quality**: We replace our InternImage-based detector with two alternatives: Faster R-CNN, which performs worse on COCO, and Co-DETR, which performs comparably on COCO.
> As shown in the following table, the accuracy drops when using Faster R-CNN (64.1→63.4), while it remains comparable when using Co-DETR (64.1→63.8), indicating that Grounded CoT benefits from stronger detectors and achieves stable performance once the detector is sufficiently strong.
>
> |Faster R-CNN (weaker)|Co-DETR (comparable)|Ours|
> |-|-|-|
> |63.4|63.8|**64.1**|
>
> **Vary OCR quality**:  We replace PP-OCRv4 with two alternatives: EasyOCR, a commonly used baseline OCR model, and AzureOCR, one of the strongest commercial OCR systems.
> As shown in the following table, the accuracy drops when using EasyOCR (64.1→62.9), while it remains comparable when using AzureOCR (64.1→64.0), indicating that Grounded CoT benefits from stronger OCR models and achieves stable performance once the OCR model is sufficiently strong.
>
> |EasyOCR (weaker)|Ours|AzureOCR (stronger)|
> |-|-|-|
> |62.9|**64.1**|64.0|

---

> ### Author Response · Authors · 2025-11-24
> **Response to Reviewer x562 (2/2)**
>
> > **Weakness 3**: The paper does not include a dedicated limitations discussion. A concise reflection on dataset scope and failure modes would improve transparency and guide responsible use.
>
> **A**: Thank you for pointing this out.
> We have now included a dedicated limitations discussion on dataset scope and remaining model errors in **Appendix F**.
>
> **Dataset scope**: Regarding the images, InfoDet focuses on infographics that combine texts, charts, and HROs, while images lacking charts or HROs were excluded during collection.
> Consequently, InfoDet is less suitable as the sole training source for applications targeting graphic designs without charts or scientific documents without HROs.
> Regarding the annotations, a model-in-the-loop method is utilized to generate annotations for the real infographics.
> This method helps maintain consistent annotation guidelines across the dataset, but it may also introduce consistent errors in ambiguous cases.
> This should be taken into account for tasks requiring highly precise detection results.
>
> **Remaining model errors**: At the end of the refinement process, we randomly sample 1,250 infographics to evaluate annotation quality.
> While the precision (93.9%) and recall (96.7%) are comparable to those of widely used object detection datasets, a small number of annotation errors still remain.
> The errors are summarized as follows.
> The most frequent errors include:
> 1) imprecise bounding-box localization due to ambiguous element boundaries;
> 2) marks or annotations mistakenly detected as HROs due to their high visual similarity; and
> 3) missed tiny elements due to their limited scale.
>
> |Error Type|Percentage|
> |-|-|
> |Imprecise bounding-box localization|36.5%|
> |Marks or annotations detected as HROs|32.9%|
> |Missed tiny elements|20.4%|
> |Others|10.2%|

---

### Official Review · Reviewer_Jyhz · 2025-11-01

**Soundness:** 3
**Presentation:** 3
**Contribution:** 3
**Rating:** 6
**Confidence:** 4

**Summary:**

This paper introduces InfoDet, a large-scale dataset for infographic element detection containing 11,264 real and 90,000 synthetic infographics with over 14 million bounding box annotations (texts, charts, HROs, sub-elements). Synthetic infographics are generated using 1,072 template-based methods, while real infographics are annotated via model-in-the-loop approach achieving 93.9% precision and 96.7% recall. The paper demonstrates usefulness through three applications: (1) Thinking-with-Boxes scheme improving o4-mini on ChartQAPro by 1.7pp, (2) comparing 11 object detection models showing foundation models struggle (DINO-X: 14.0 chart AP) while Co-DETR achieves 88.2 mAP after InfoDet fine-tuning, and (3) transfer learning on Rico and DocGenome showing InfoDet pre-training improves performance.

**Strengths:**

**1. Comprehensive and Well-Designed Dataset**
- First large-scale infographic dataset (101,264 samples vs. prior Borkin et al. 393 samples) strategically combining real and synthetic data for authenticity and scalability
- Efficient model-in-the-loop annotation achieving quality comparable to COCO (precision 93.9%, recall 96.7% vs. COCO's 71.9%/83.0%)
- Multi-level annotations: element-level (charts, HROs) and mark-level (26 sub-element categories) providing fine-grained labels
- Verified diversity across 67 chart types, 96 topics, 15 visual styles (Appendix F)

**2. Important Empirical Findings on Model Capabilities**
- **Foundation model limitations**: State-of-the-art models like DINO-X achieve only 14.0 chart AP and 15.0 HRO AP in zero-shot, with few-shot prompting showing minimal improvement (MQ-GLIP 4-shot: 16.2 chart AP). This reveals natural scene pre-training is insufficient for graphics domain.
- **Fine-tuning effectiveness**: Co-DETR improves dramatically from 27.6 AP (4-shot) to 88.2 AP (InfoDet training), demonstrating value of large-scale quality data
- **Generalization capability**: Cross-dataset evaluation shows InfoDet -> VG-DCU transfer drops only 17.1% mAP vs. 53.7% drop in reverse direction, indicating infographic data learns more general representations than plain charts

**3. Thorough Experimental Protocol and Transparent Reporting**
- Standardized training protocol ensuring fair comparison across 11 models
- Ablation studies demonstrating complementary roles of visual and textual prompts (64.1 combined vs. 62.8 visual-only, 61.6 textual-only)
- Transfer learning validated on Rico (42.1 -> 50.6 -> 53.6) and DocGenome (69.0 -> 74.4 -> 80.0)
- Honest reporting of limitations (minimal improvement on plain single charts, o3's instruction-following issues)

**Weaknesses:**

**1. Dataset Construction Issues: Representativeness and Transparency**
- **Fine-grained annotation imbalance**: 75 chart types exist only for synthetic infographics. Authors mention GPT-4o achieved only 61.49% accuracy on real infographics but provide no alternative approach (human annotation? better models?), leaving dataset incomplete.
- **Annotation process opaque**: No information on expert demographics (number? background: medical imaging experts? graphic designers? CV researchers?), training process, or inter-annotator agreement. "Multiple rounds of refinement" lacks specifics: how many rounds? How did precision/recall improve each round? What was initial auto-annotation quality? These details are critical for reproducibility and trustworthiness.

**2. Limited Application Improvements with Insufficient Analysis**
- **Grounded CoT modest gains**: Overall 1.7pp improvement is limited, especially negligible on plain single charts (58.1→60.6) with clear improvement only on infographic multiple (70.6→72.5), suggesting method effectiveness is confined to complex scenarios.
- **Detection evaluation lacks depth**: No analysis of post-fine-tuning regression on other scenarios (natural images). Mark-level performance drops 6.5pp from element-level (76.1% -> 69.6%) without investigating causes (more classes? smaller objects? ambiguous boundaries?). Chart detection improves 3.2× (27.6 -> 88.2) while HRO only 2.5× (25.5 -> 64.0) - why is HRO harder? Which HRO types (data-related vs. theme-related) are particularly challenging?

**3. Insufficient Motivation and Missing Broader Context**
- **Limited practical justification**: Introduction only states "charts are fundamental" without explaining why infographic understanding matters or concrete use cases (e.g., social media analysis, news understanding, educational accessibility, automated fact-checking for visually impaired users). Lack of real-world deployment scenarios weakens motivation.
- **Limited generalization evaluation**: Grounded CoT only tested on ChartQAPro - does it work on other chart QA benchmarks (ChartQA, PlotQA)? Transfer learning only on Rico/DocGenome - what about other layout detection datasets?

**Questions:**

**1. Dataset Quality: Synthetic Data and Annotation Process**
- How does the 8:1 synthetic-to-real ratio with observed biases (hand-drawn style overrepresentation, missing composite charts) affect learned model behavior?
- Why is fine-grained chart type annotation impossible for real infographics? Were better models (Gemini 2.5 pro, GPT5, Claude Sonnet 4) or human annotation attempted beyond GPT-4o's 61.49%?
- How many expert annotators participated with what backgrounds and training? How many refinement rounds occurred with what precision/recall progression? What was initial auto-annotation quality and inter-annotator agreement, especially for ambiguous chart-HRO boundaries?

**2. Application Mechanisms and Effectiveness**
- For grounded CoT: Aren't bbox coordinates sufficient instead of full content text? Does current approach constitute "answer leakage" via OCR? How does bbox detection error (~12% given 88.2 AP) impact final QA performance - correlation analysis available?
- Why minimal improvement on plain single charts - is native VLM visual reasoning already sufficient there?
- For object detection: Does InfoDet fine-tuning cause regression on other domains (natural images)? What causes 6.5pp mark-level drop - class count, object size, or boundary ambiguity? Root cause of foundation model failure - simply pre-training data mismatch or intrinsic task difficulty?

**3. Evaluation Rigor and Generalization**
- Does grounded CoT generalize to other chart QA benchmarks (ChartQA, PlotQA)?
- Does transfer learning generalize to other layout detection datasets beyond Rico/DocGenome?
- Which specific component types in Rico/DocGenome benefit most from InfoDet pre-training, and why larger improvement on DocGenome?

---

> ### Author Response · Authors · 2025-11-24
> **Response to Reviewer Jyhz (1/6)**
>
> We sincerely appreciate your valuable and helpful comments.
> Per your suggestions, we have carefully addressed the identified weaknesses.
> Please see our response below.
>
> > **Weakness 1.1**: Fine-grained annotation imbalance: 75 chart types exist only for synthetic infographics. Authors mention GPT-4o achieved only 61.49% accuracy on real infographics but provide no alternative approach (human annotation? better models?), leaving dataset incomplete.
>
> > **Question 1.2**: Why is fine-grained chart type annotation impossible for real infographics? Were better models (Gemini 2.5 pro, GPT5, Claude Sonnet 4) or human annotation attempted beyond GPT-4o's 61.49%?
>
> **A**: To complement the dataset with chart-type annotations, we have attempted two more methods:
> 1) GPT-5; and
> 2) a two-stage CLIP-based linear probing method that first predicts one of 10 coarse categories (e.g., bar, line) and then refines it into one of 75 chart types using classifiers trained on our synthetic infographics.
>
> GPT-5 achieves an accuracy of 69.64%, while the two-stage method achieves 78.80%.
> We have now provided the chart-type annotations generated using the two-stage method on Hugging Face.
> The moderate accuracy reflects the inherent difficulty of classifying 75 fine-grained chart types, in which some categories are visually very similar (e.g., gauge vs. donut charts), making fully reliable annotation of real infographics impossible without expert verification.
> The authors are currently manually verifying the annotations, and the verified annotations will be provided in the final version.
> We have clarified this in **Appendix D**.

---

> ### Author Response · Authors · 2025-11-24
> **Response to Reviewer Jyhz (2/6)**
>
> > **Weakness 1.2**: Annotation process opaque: No information on expert demographics (number? background: medical imaging experts? graphic designers? CV researchers?), training process, or inter-annotator agreement. "Multiple rounds of refinement" lacks specifics: how many rounds? How did precision/recall improve each round? What was initial auto-annotation quality? These details are critical for reproducibility and trustworthiness.
>
> > **Question 1.3**: How many expert annotators participated with what backgrounds and training? How many refinement rounds occurred with what precision/recall progression? What was initial auto-annotation quality and inter-annotator agreement, especially for ambiguous chart-HRO boundaries?
>
> **A**: We have now provided additional clarifications on the expert background, refinement rounds, and the inter-annotator agreement in **Appendix E**.
>
> **Expert background**: Seven experts participated in the annotation process.
> Two are Ph. D. students majoring in data visualization who are co-authors of this paper.
> The other five are experienced annotators from a vendor company, each with at least two years of experience in object detection annotation, practical experience in plain chart annotation, and at least a bachelor’s degree.
>
> **Refinement rounds**: The annotation refinement process spanned four months, with one round lasting two weeks, resulting in a total of eight rounds.
> Each round followed this procedure:
> 1) The Ph.D. experts reviewed existing annotations to identify potential issues;
> 2) They trained the five annotators to perform targeted corrections;
> 3) The detection model was fine-tuned on the corrected annotations to improve subsequent predictions.
>
> Since evaluating annotation precision and recall requires additional human effort, we evaluated precision and recall at three stages: before refinement, after the fourth round, and after all rounds.
> As shown in the following table, the precision and recall steadily improved.
>
> |Refinement Round|Precision|Recall|
> |-|-|-|
> |Before the refinement|69.6%|77.0%|
> |After the fourth refinement round|89.0%|91.3%|
> |After all refinement rounds|93.9%|96.7%|
>
> **Inter-annotator agreement**: To ensure high inter-annotator agreement in the refinement rounds, the five annotators were trained each round by the two Ph.D. experts, and their corrections were required to reach an average IoU of 0.95 before annotation.
> To further verify the consistency between the Ph.D. experts, we conducted an inter-annotator agreement study on 1,000 real infographics.
> For each infographic, we first generated annotations using the model before refinement, then randomly selected one element from those annotations.
> The Ph.D. experts independently corrected the element, adjusting its location and category as needed.
> They were also asked to mark the occurrences of ambiguous chart–HRO boundaries.
> The results on all elements and on 39 elements with ambiguous chart-HRO boundaries are as follows.
> The experts performed consistent corrections (average IoU=0.95, category agreement=0.99).
> These corrections differed from the model predictions, which showed lower agreement (average IoU=0.90, category agreement=0.96), indicating that they can effectively mitigate systematic biases in the model annotations.
> Even for the elements with ambiguous boundaries, the agreement remains high (average IoU=0.84, category agreement=0.92), further confirming the reliability of the corrections.
>
> |All Elements|Average IoU|Category Agreement|
> |-|-|-|
> |Expert-Expert|**0.95**|**0.99**|
> |Expert-Model|0.90|0.96|
>
> |Elements with Ambiguous Boundaries|Average IoU|Category Agreement|
> |-|-|-|
> |Expert-Expert|**0.84**|**0.92**|
> |Expert-Model|0.77|0.91|

---

> ### Author Response · Authors · 2025-11-24
> **Response to Reviewer Jyhz (3/6)**
>
> > **Weakness 2.1**: Grounded CoT modest gains: Overall 1.7pp improvement is limited, especially negligible on plain single charts (58.1→60.6) with clear improvement only on infographic multiple (70.6→72.5), suggesting method effectiveness is confined to complex scenarios.
>
> > **Question 2.2**: Why minimal improvement on plain single charts - is native VLM visual reasoning already sufficient there?
>
> **A**: The modest overall performance gain of Grounded CoT on the ChartQAPro benchmark is due to the inclusion of both plain and infographic charts.
> Questions on plain charts typically only require reading a label or number, and therefore require less visual grounding and reasoning.
> In these cases, current VLMs already achieve near-saturated performance.
> As a result, our method performs similarly to the baselines on these questions, which lowers the overall performance gain observed across the full benchmark.
> Grounded CoT is specifically designed to help in scenarios where the answer requires reasoning over grounded visual elements.
> These characteristics appear primarily in infographic charts in chartQAPro.
> To better illustrate the benefit of our method on infographic charts, we compare it with the direct prompting baseline on infographic charts only, using models o1, o3, and o4-mini.
> The results show that our method achieves clear and consistent gains across all models on infographic charts (+4.6\% on o1, +2.8\% on o3, and +1.7\% on o4-mini), demonstrating its effectiveness in handling complex visual reasoning tasks.
>
> |Model|Direct|Grounded CoT|$\Delta$|
> |-|-|-|-|
> |o1|66.1|70.7|4.6|
> |o3|65.7|68.5|2.8|
> |o4-mini|69.6|71.3|1.7|
>
> > **Weakness 2.2**: Detection evaluation lacks depth: No analysis of post-fine-tuning regression on other scenarios (natural images). Mark-level performance drops 6.5pp from element-level (76.1% -> 69.6%) without investigating causes (more classes? smaller objects? ambiguous boundaries?). Chart detection improves 3.2× (27.6 -> 88.2) while HRO only 2.5× (25.5 -> 64.0) - why is HRO harder? Which HRO types (data-related vs. theme-related) are particularly challenging?
>
> > **Question 2.3**: For object detection: Does InfoDet fine-tuning cause regression on other domains (natural images)? What causes 6.5pp mark-level drop - class count, object size, or boundary ambiguity? Root cause of foundation model failure - simply pre-training data mismatch or intrinsic task difficulty?
>
> **A**: Thank you for pointing this out.
> Below, we provide additional analyses of the evaluation results.
> These analyses are also incorporated in **Appendix K.2**.
>
> **Post-fine-tuning regression on natural images**: Our object detection model is built by fine-tuning InternImage-L with the DINO detector.
> After fine-tuning, the model no longer predicts natural-image classes, making direct evaluation on COCO or other natural-image benchmarks infeasible.
> Developing a unified model that robustly handles both infographics and natural images would indeed be highly beneficial, but it is beyond the scope of our current work, which focuses exclusively on understanding infographics.
>
> **Mark-level performance drop**: The drop in detection performance from element-level to mark-level is primarily due to two factors:
> 1) The mark-level annotations contain more classes with high visual similarity (e.g., "gauge\_mark" and "donut\_mark"), which increases classification difficulty; and
> 2) The mark-level annotations contain smaller chart sub-elements, which makes localization harder.
>
> Boundary ambiguity is not a critical factor because most mark-level categories correspond to clear geometric shapes with well-defined boundaries.
>
> **Why HRO detection is harder**:
> Analysis of the remaining errors of our InternImage-based model in Appendix F reveals that the most frequent errors include:
> 1) imprecise bounding-box localization due to ambiguous element boundaries (36.5%);
> 2) marks or annotations mistakenly detected as HROs due to their high visual similarity (32.9%); and
> 3) missed tiny elements due to their limited scale (20.4%).
>
> The first and third errors occur more often for HROs, while the second error occurs only for HROs, making them harder to detect.
>
> **Which HRO type is more challenging**:
> Data-related HROs are more challenging to detect for three reasons:
> 1) They are more likely to be closely integrated with charts, resulting in ambiguous boundaries;
> 2) Marks and annotations are more likely to be mistakenly detected as data-related HROs; and
> 3) They are usually smaller and harder to localize.
>
> **Root cause of foundation model failure**:
> While foundation models achieve limited performance (mAP < 20), traditional models fine-tuned on our dataset can achieve much higher performance (mAP > 75) despite weaker inherent capabilities.
> This suggests that the primary cause of foundation model failure is pre-training data mismatch, rather than the intrinsic difficulty of the task.

---

> ### Author Response · Authors · 2025-11-24
> **Response to Reviewer Jyhz (4/6)**
>
> > **Weakness 3.1**: Limited practical justification: Introduction only states "charts are fundamental" without explaining why infographic understanding matters or concrete use cases (e.g., social media analysis, news understanding, educational accessibility, automated fact-checking for visually impaired users). Lack of real-world deployment scenarios weakens motivation.
>
> **A**:
> We have now strengthened the **Introduction** by adding more concrete real-world applications.
> In particular, we highlight that infographic understanding using VLMs has become
> increasingly important in scenarios such as news analysis [1] and accessible content generation [2].
> These applications rely heavily on the accurate grounding of infographic elements.
> However, existing VLMs still struggle with this capability, which directly motivates our work.
>
> > **Weakness 3.2**: Limited generalization evaluation: Grounded CoT only tested on ChartQAPro - does it work on other chart QA benchmarks (ChartQA, PlotQA)? Transfer learning only on Rico/DocGenome - what about other layout detection datasets?
>
> > **Question 3.1**: Does grounded CoT generalize to other chart QA benchmarks (ChartQA, PlotQA)?
>
> > **Question 3.2**: Does transfer learning generalize to other layout detection datasets beyond Rico/DocGenome?
>
> **A**: We have now conducted additional generalization evaluation for Grounded CoT and transfer learning.
>
> **Grounded CoT**: We have now compared Grounded CoT against direct prompting, the best-performing baseline, on ChartQA and PlotQA using o1.
> The results are as follows.
> Our method performs better than direct prompting on both benchmarks, indicating that its benefits extend beyond ChartQAPro.
> We have included these results in **Appendix J.1**.
>
> |Method|ChartQA|PlotQA|
> |-|-|-|
> |Direct|75.6|30.6|
> |Grounded CoT (ours)|**79.2**|**31.3**|
>
> **Transfer learning**: We have now applied our model to PosterLayout [3], which contains 9,974 posters annotated with bounding boxes of text, logo, and underly.
> The results are as follows.
> Pre-training on InfoDet improves model performance when fine-tuned on
> PosterLayout, both on its own (62.5→73.6) and when combined with existing large-scale pre-training data (74.9→76.0).
> We have included these results in **Sec. 4.3.2**.
>
> |Pre-Training|PosterLayout|
> |-|-|
> |-|62.5|
> |InfoDet|73.6|
> |ImageNet-22K, Objects365, COCO|74.9|
> |ImageNet-22K, Objects365, COCO, InfoDet|**76.0**|

---

> ### Author Response · Authors · 2025-11-24
> **Response to Reviewer Jyhz (5/6)**
>
> > **Question 1.1**: How does the 8:1 synthetic-to-real ratio with observed biases (hand-drawn style overrepresentation, missing composite charts) affect learned model behavior?
>
> **A**: Although the synthetic infographics contain more hand-drawn style charts and composite charts appear only in real infographics, we do not observe negative effects on model behavior.
> Instead, the model performs well on both hand-drawn and composite charts, effectively learning from both sources despite the 8:1 synthetic-to-real ratio.
> This benefit comes from combining complementary strengths: the synthetic infographics provide controlled variation and scalability, while the real infographics capture authentic design practices.
> We believe that systematically quantifying how mixing data with different characteristics affects model behavior is a promising direction for future research and deserves a separate publication.
>
> > **Question 2.1**: For grounded CoT: Aren't bbox coordinates sufficient instead of full content text? Does current approach constitute "answer leakage" via OCR? How does bbox detection error (~12% given 88.2 AP) impact final QA performance - correlation analysis available?
>
> **A**: Thank you for pointing this out.
> Below, we provide clarifications regarding potential answer leakage and the necessity of full content text, and the impact of detection quality on final QA performance.
>
> **Potential answer leakage and the necessity of full content text**:
> We believe that providing content text does not constitute answer leakage for two reasons.
> First, the evaluated VLMs (o1, o3, and o4-mini) have strong OCR capabilities and can read the same text directly from the image. Therefore, providing this text does not give the model information it could not obtain.
> Second, we do not indicate which pieces of text are relevant to the answer.
> Instead, we uniformly extract all textual content within the infographic.
> Compared to using only bounding-box coordinates, providing full text reduces the model’s effort on low-level text parsing and allows it to focus on reasoning over the detected elements, thereby better leveraging its visual reasoning capability.
>
> **Impact of detection quality**: We evaluate the impact of detection quality by replacing our InternImage-based detector with two alternatives: Faster R-CNN, which performs worse on COCO, and Co-DETR, which performs comparably on COCO.
> As shown in the following table, the accuracy drops when using Faster R-CNN (64.1→63.4), while it remains comparable when using Co-DETR (64.1→63.8), indicating that grounded CoT benefits from stronger detectors and achieves stable performance once the detector is sufficiently strong.
>
> |Faster R-CNN (weaker)|Co-DETR (comparable)|Ours|
> |-|-|-|
> |63.4|63.8|**64.1**|
>
> > **Question 3.3**: Which specific component types in Rico/DocGenome benefit most from InfoDet pre-training, and why larger improvement on DocGenome?
>
> **A**: We have now evaluated the AP changes of Rico and DocGenome component types when fine-tuned from scratch versus pre-trained on InfoDet.
> The top three component types benefiting most from InfoDet pre-training are:
>
> - Rico: Date Picker (+29.5), Slider (+17.4), Radio Button (+16.6)
>
> - DocGenome: Text (+11.0), Code (+8.5), Title (+7.6)
>
> The full results are provided in **Appendix K.2**.
> The improvement on Rico (42.1→50.6, $\Delta$=8.5) is larger than that on DocGenome (69.0→74.4, $\Delta$=5.4).
> This is because Rico contains more complex UI hierarchies, whereas DocGenome components have nearly no hierarchical structure.
> The hierarchical structures in Rico are more similar to InfoDet’s chart and sub-element hierarchies, allowing the model to better leverage the pre-training.

---

> ### Author Response · Authors · 2025-11-24
> **Response to Reviewer Jyhz (6/6)**
>
> ## References
>
> [1] Yuwei Chen and Ming-Ching Chang. Transformer based image-text consistency analysis for infographic articles. In IEEE International Conference on Multimedia Information Processing and Retrieval, pp. 47–52, 2023.
>
> [2] Clarissa Hohenwalde and Antonios Hazim. Enhancing graphic accessibility for blind and visually impaired people: A systematic comparison of vision language models (VLM). INFORMATIK, pp. 613, 2025.
>
> [3] Hsiao Yuan Hsu, Xiangteng He, Yuxin Peng, Hao Kong, and Qing Zhang. Posterlayout: A new benchmark and approach for content-aware visual-textual presentation layout. In Proceedings of the IEEE/CVF Conference on Computer Vision and Pattern Recognition, pp. 6018–6026, 2023.

---

### Official Review · Reviewer_WrQm · 2025-11-02

**Soundness:** 3
**Presentation:** 3
**Contribution:** 2
**Rating:** 4
**Confidence:** 3

**Summary:**

This paper presents InfoDet, a large-scale dataset for infographic-style element detection comprising about 101K images and 14M bounding boxes, annotated over text, charts, human-recognizable objects, and fine-grained chart components (26 mark types, 75 chart types). The authors argue that many failures of current multimodal/chart QA systems arise from inadequate grounding in cluttered infographic layouts, and they show that supplying detected elements to recent vision–language models improves performance on challenging multi-chart benchmarks.

**Strengths:**

1.The paper targets a real and under-served pain point in multimodal/chart understanding—current VLMs and chart/infographic QA systems often fail not because they cannot “reason,” but because they cannot reliably ground the relevant regions in cluttered infographic-style inputs.

2.The proposed InfoDet dataset is both large (≈101K images, ≈14M boxes) and unusually fine-grained, covering text, charts, and human-recognizable objects (HROs), as well as 26 chart-level marks and 75 chart types. This level of structural annotation is rare in existing public datasets and is directly useful for detection-then-reasoning pipelines.

3.The “thinking-with-boxes” usage demonstrates that plugging the detected elements into strong VLMs yields consistent gains on challenging, multi-chart settings (e.g., ChartQAPro), which shows the dataset is not only a collection effort but can translate into practical improvements in multimodal QA.

**Weaknesses:**

1.The core novelty lies in building a large, high-quality dataset and a reasonable model-in-the-loop pipeline, plus a demonstrative prompting scheme. Compared to typical ICLR work, the methodological/learning novelty is modest.

2. Given that the dataset provides structured and layered infographic elements, the paper could reasonably be expected to propose a model or training scheme that explicitly exploits this structure (for example, through element-level selection, layout-aware fusion, or hierarchical grounding). The current usage mainly demonstrates utility through prompting, rather than through a dedicated model design that fully leverages the annotations.

**Questions:**

None

---

> ### Author Response · Authors · 2025-11-24
> **Response to Reviewer WrQm**
>
> We sincerely appreciate your valuable and constructive comments.
> Per your suggestions, we have carefully addressed the identified weaknesses. Please see our response below.
>
> > **Weakness 1**: The core novelty lies in building a large, high-quality dataset and a reasonable model-in-the-loop pipeline, plus a demonstrative prompting scheme. Compared to typical ICLR work, the methodological/learning novelty is modest.
>
> **A**:
> Our submission is to the Datasets and Benchmarks area, where novelty is expected to come primarily from the dataset. annotation method, and evaluation protocol, rather than proposing a new learning architecture.
> In this sense, our work offers three technical contributions:
>
> 1. A large-scale dataset for infographic element detection.
> Our dataset includes 11,264 real and 90,000 synthetic infographics with over 14 million bounding-box annotations, making it the largest dataset for infographic element detection to date.
> It also includes a test set of 5,000 infographics with manually refined annotations, enabling reliable evaluation of detection performance.
>
> 2. An effective and cost-efficient model-in-the-loop annotation pipeline.
> We design a model-in-the-loop pipeline that co-develops an object detection model and the annotation process, enabling the model and the annotations to iteratively improve together.
> This pipeline yields annotations with a precision of 93.9% and a recall of 96.7%, comparable to those of widely used object detection datasets.
> It also reduces human annotation cost from more than $\textdollar$30,000 to $\textdollar$3,000.
>
> 3. A Thinking-with-Boxes scheme for VLM reasoning.
> Our Thinking-with-Boxes scheme provides grounded annotations of texts, charts, and HROs to guide VLMs toward step-by-step reasoning.
> Our method achieves clear and consistent gains on infographic charts in the ChartQAPro benchmark (+4.6% on o1, +2.8% on o3, and +1.7% on o4-mini), demonstrating its effectiveness in handling complex visual reasoning tasks.
>
> In summary, while we do not claim major learning novelty, we believe the above contributions align well with the expectations of the Datasets and Benchmarks area.
> Considering the recent progress on infographic understanding (e.g., Gemini 3 Pro) and generation (e.g., Nano Banana Pro), we believe that our dataset will provide support to advance future methodological and learning innovations.
>
> > **Weakness 2**: Given that the dataset provides structured and layered infographic elements, the paper could reasonably be expected to propose a model or training scheme that explicitly exploits this structure (for example, through element-level selection, layout-aware fusion, or hierarchical grounding). The current usage mainly demonstrates utility through prompting, rather than through a dedicated model design that fully leverages the annotations.
>
> **A**:
> Per your suggestion, we have made an initial effort toward a hierarchy-guided matching (HGM) scheme that leverages the hierarchical relationships between charts and their sub-elements in the annotations.
> We introduce this method in **Appendix I.3**.
> Specifically, in the DINO detector, Hungarian matching determines one-to-one correspondences between predictions and ground-truth boxes for computing the training loss.
> Previous works such as DN-DETR [1] and Stable-DINO [2] have shown that a reliable matching process is crucial for stable model optimization.
> Motivated by this, we aim to incorporate hierarchy constraints into the matching process, ensuring that chart sub-elements are contained within their charts.
> However, directly incorporating such constraints turns the matching into a quadratic assignment problem, which is NP-hard and computationally infeasible.
> Therefore, we adopt a two-stage matching method, first performing standard Hungarian matching and then refining the matching iteratively to satisfy the hierarchy constraints.
> We have integrated this method into the graphic layout detection experiments in **Sec. 4.3**.
> The results are as follows.
> Incorporating HGM increases the gain of InfoDet pre-training from 1.8 to 2.6 for Rico, 1.3 to 2.1 for DocGenome, and from 1.1 to 1.5 for PosterLayout, suggesting that exploiting the hierarchical structure in InfoDet is indeed beneficial and a promising direction for future research.
>
> |Pre-Training|Rico|DocGenome|PosterLayout|
> |-|-|-|-|
> |ImageNet-22K, Objects365, COCO|51.8|78.7|74.9|
> |ImageNet-22K, Objects365, COCO, InfoDet|53.6|80.0|76.0|
> |ImageNet-22K, Objects365, COCO, InfoDet w. HGM|**54.4**|**80.8**|**76.4**|
>
> **References**
>
> [1] Feng Li, Hao Zhang, Shilong Liu, Jian Guo, et al. DN-DETR: Accelerate DETR training by introducing query denoising. In Proceedings of the IEEE/CVF conference on computer vision and pattern recognition, pp. 13619–13627, 2022.
>
> [2] Shilong Liu, Tianhe Ren, Jiayu Chen, Zhaoyang Zeng, et al. Detection transformer with stable matching. In Proceedings of the IEEE/CVF International Conference on Computer Vision, pp. 6491–6500, 2023.

---

### Official Review · Reviewer_fMwe · 2025-11-05

**Soundness:** 4
**Presentation:** 4
**Contribution:** 3
**Rating:** 6
**Confidence:** 4

**Summary:**

This paper introduces InfoDet, a large-scale dataset for infographic element detection, designed to improve visual grounding and chart understanding in vision-language models (VLMs). The dataset consists of 11,264 real and 90,000 synthetic infographics, totaling over 14 million bounding box annotations covering texts, charts, human-recognizable objects (HROs), and sub-elements like bars and legends. The authors develop a hybrid annotation pipeline combining model-in-the-loop and programmatic methods to efficiently produce high-quality labels. Three key applications are demonstrated: (1) a Thinking-with-Boxes scheme that enhances grounded reasoning in VLMs , (2) benchmarking 11 object detection models on infographic data, and (3) transferring InfoDet-trained models to graphic layout detection in documents and UI design.

**Strengths:**

The paper’s methodological rigor and practical significance are outstanding. The hybrid annotation strategy—combining synthetic and real data under a model-in-the-loop refinement—balances scalability and accuracy, achieving dataset quality comparable to COCO. The scale (over 100k infographics) and fine-grained annotations (charts, HROs, sub-elements) fill a clear research gap. The grounded CoT prompting demonstrates genuine insight into how structured visual grounding improves reasoning in modern VLMs. Empirical evaluations are extensive and carefully analyzed, revealing key insights (e.g., why zero-/few-shot detection fails on infographics). Finally, ethical considerations and licensing transparency are meticulously handled.

**Weaknesses:**

First, the paper lacks quantitative validation for synthetic data fidelity and bias, which is a critical limitation given that nearly 90% of InfoDet consists of synthetic infographics. Without quantitative analysis or cross-domain alignment metrics—such as style distribution, semantic bias, or embedding similarity—the fidelity of synthetic data relative to real samples remains uncertain, casting doubt on downstream generalization and fairness.
Second, there is insufficient analysis of annotation tradeoffs and reproducibility. Although the model-in-the-loop annotation pipeline is innovative, the authors do not quantify annotation cost, human effort, or refinement rounds, nor do they analyze how annotation quality affects detection accuracy, which weakens the dataset’s reproducibility and practical replicability.
Third, the evaluation of the Thinking-with-Boxes method is limited in scope and validation. The experiments are restricted to a single benchmark (ChartQAPro) with relatively small gains (around 3–5%) and no statistical or qualitative error analysis, suggesting that the improvements may stem from prompt design rather than genuine reasoning enhancement.
Finally, there is a lack of formal evaluation of annotation consistency and labeling noise. The reported annotation precision and recall are derived from only 1,250 samples and omit inter-annotator agreement or systematic error analysis. This absence of consistency validation leaves the internal reliability of the dataset uncertain and may affect the credibility of subsequent benchmarking results.

**Questions:**

How does annotation quality vary across infographic types (e.g., dense textual vs. image-heavy layouts)?
Could the Thinking-with-Boxes approach extend to document reasoning tasks beyond charts (e.g., scientific figures, tables)?
Have the authors considered releasing segmentation masks or relation graphs between detected elements to support compositional reasoning?
Could the authors quantify the annotation process in terms of human effort and computational
What are the computational costs of fine-tuning large detection models (e.g., Co-DETR) on InfoDet relative to standard datasets?

---

> ### Author Response · Authors · 2025-11-24
> **Response to Reviewer fMwe (1/4)**
>
> We sincerely appreciate your valuable and helpful suggestions.
> Per your suggestions, we have carefully addressed the identified weaknesses. Please see our response below.
>
> > **Weakness 1**: First, the paper lacks quantitative validation for synthetic data fidelity and bias, which is a critical limitation given that nearly 90% of InfoDet consists of synthetic infographics. Without quantitative analysis or cross-domain alignment metrics—such as style distribution, semantic bias, or embedding similarity—the fidelity of synthetic data relative to real samples remains uncertain, casting doubt on downstream generalization and fairness.
>
> **A**: Per your suggestion, we have now provided quantitative analysis of synthetic data fidelity and bias in **Appendix H**.
>
> **Data fidelity**: We measure embedding similarity by evaluating how well the synthetic infographics cover the real ones.
> Specifically, we extract the embeddings of all infographics using CLIP and project them into a two-dimensional space using UMAP [1].
> The space is divided into a uniform grid, and a real infographic is considered "covered" if at least one synthetic infographic falls in the same grid cell.
> The results show that the distributions of real and synthetic infographics largely overlap, with 92.64% of real infographics covered by synthetic ones, indicating high similarity.
>
> We also compare the distributions of four key attributes: chart type, infographic topic, visual style, and communication tone.
> These attributes are extracted using Gemini-2.5-flash, and their alignment is measured with Jensen-Shannon divergence [2], a commonly used measure of similarity between probability distributions.
> The results are as follows.
> The divergences for infographic topics (0.053), visual styles (0.022), and communication tones (0.038) are low, suggesting strong alignment.
> The divergence for chart types (0.149) is relatively higher because the synthetic infographics cover all chart types and include a higher proportion of rare ones (e.g., circular bar charts, waffle charts) to ensure sufficient coverage.
>
> |Attribute|JS Distance|
> |-|-|
> |Chart Type|0.149|
> |Infographic Topic|0.053|
> |Visual Style|0.022|
> |Communication Tone|0.038|
>
> **Data bias**: To identify potential biases, we compute pointwise mutual information (PMI) [3], a standard metric for measuring co-occurrence strength, across attribute pairs.
> We have manually inspected all pairs with PMIs greater than 1 and found that these high-PMI pairs (e.g., environment topic + concerned tone) align with common communication patterns and do not indicate harmful or misleading bias.
> This confirms that the synthetic infographics exhibit no evident harmful bias.
>
> Together, these quantitative analyses provide evidence that the synthetic infographics closely mirror real ones, supporting their use for downstream generalization without introducing harmful bias.

---

> ### Author Response · Authors · 2025-11-24
> **Response to Reviewer fMwe (2/4)**
>
> > **Weakness 2**: Second, there is insufficient analysis of annotation tradeoffs and reproducibility. Although the model-in-the-loop annotation pipeline is innovative, the authors do not quantify annotation cost, human effort, or refinement rounds, nor do they analyze how annotation quality affects detection accuracy, which weakens the dataset’s reproducibility and practical replicability.
>
> **A**: We have now provided additional clarifications regarding annotation cost and human effort, the refinement process, and how annotation quality relates to detection performance in **Appendix E**.
>
> **Annotation cost and human effort**: The refinement rounds required around $\textdollar$3,000 in human annotation cost and 313 GPU-hours for model fine-tuning on an NVIDIA Tesla V100, priced at approximately $\textdollar$0.7 per GPU-hour.
> In total, the refinement process cost roughly $\textdollar$3,220.
> In contrast, fully manual annotation of all real infographics is estimated to cost more than $\textdollar$30,000, due to the expertise required and the need for multi-round quality verification to ensure annotation accuracy.
> Therefore, our model-in-the-loop method offers a cost-efficient alternative while maintaining high annotation quality.
> It also produces a model that can generate reliable annotations for future chart-related datasets, thereby reducing annotation costs.
>
> **Refinement process**: The annotation refinement process spanned four months, with one round lasting two weeks, resulting in a total of eight rounds.
> Seven experts participated in the annotation process.
> Two are Ph. D. students majoring in data visualization who are co-authors of this paper.
> The other five are experienced annotators from a vendor company, each with at least two years of experience in object detection annotation, practical experience in plain chart annotation, and at least a bachelor’s degree.
> Each round followed this procedure:
> 1) The Ph.D. experts reviewed existing annotations to identify potential issues;
> 2) They trained the remaining annotators to perform targeted corrections;
> 3) The detection model was fine-tuned on the corrected annotations to improve subsequent predictions.
>
> We evaluated the precision and recall at three stages: before refinement, after the fourth round, and after all rounds.
> As shown in the following table, the precision and recall steadily improved.
>
> |Refinement Round|Precision|Recall|
> |-|-|-|
> |Before the refinement|69.6%|77.0%|
> |After the fourth refinement round|89.0%|91.3%|
> |After all refinement rounds|93.9%|96.7%|
>
> **Relation between annotation quality and detection accuracy**: Detection accuracy consistently improves as annotation quality increases across refinement rounds.
> Corrections in the annotations are reflected in the model output after fine-tuning, leading to higher detection accuracy.

---

> ### Author Response · Authors · 2025-11-24
> **Response to Reviewer fMwe (3/4)**
>
> > **Weakness 3**: Third, the evaluation of the Thinking-with-Boxes method is limited in scope and validation. The experiments are restricted to a single benchmark (ChartQAPro) with relatively small gains (around 3–5%) and no statistical or qualitative error analysis, suggesting that the improvements may stem from prompt design rather than genuine reasoning enhancement.
>
> **A**: We have now evaluated our Thinking-with-Boxes method on additional chart QA benchmarks and conducted both statistical and qualitative analysis.
>
> **Additional benchmarks**: To test the generalizability beyond ChartQAPro, we compared our method against direct prompting, the best-performing baseline, on ChartQA [4] and PlotQA [5] using o1.
> The results are as follows.
> Our method consistently performs better than direct prompting, indicating that its benefits extend beyond ChartQAPro.
> We have included these results in **Appendix J.1**.
>
> |Method|ChartQA|PlotQA|
> |-|-|-|
> |Direct|75.6|30.6|
> |Grounded CoT (ours)|**79.2**|**31.3**|
>
> **Statistical analysis**: To assess the robustness of the improvement, we conducted a repeated set of runs on o3, the model with the smallest improvement.
> The results are as follows.
> In both runs, our method achieves the highest accuracy among all prompting methods.
> Even when accounting for variance, its mean performance remains clearly above the baselines.
> We have included this analysis in **Appendix J.2**.
>
> |Run|Direct|CoT|PoT|Grounded CoT (ours)|
> |-|-|-|-|-|
> |The first run|60.6|60.0|59.5|**61.6**|
> |The second run|61.4|60.1|61.0|**62.3**|
> |Mean ± Std|61.0 ± 0.5|60.0 ± 0.1|60.3 ± 1.0|**62.0 ± 0.5**|
>
> **Qualitative analysis**: We have conducted qualitative analysis to show how the grounded annotations elicit the model’s reasoning capabilities in **Appendix J.3**.
> Direct prompting often leads to reliance on superficial visual cues, such as picking a label based on its position or misinterpreting relative values.
> In contrast, our method allows the model to correctly identify relevant elements and accurately compare them.
> These results indicate that our method goes beyond mere prompt design and truly leverages the model’s visual reasoning capabilities, reducing errors from reliance on superficial visual cues.
>
> > **Weakness 4**: Finally, there is a lack of formal evaluation of annotation consistency and labeling noise. The reported annotation precision and recall are derived from only 1,250 samples and omit inter-annotator agreement or systematic error analysis. This absence of consistency validation leaves the internal reliability of the dataset uncertain and may affect the credibility of subsequent benchmarking results.
>
> **A**: We have now conducted inter-annotator agreement and systematic error analysis.
>
> **Inter-annotator agreement**:
> To ensure high inter-annotator agreement in the refinement rounds, the five annotators were trained each round by the two Ph.D. experts, and their corrections were required to reach an average IoU of 0.95 before annotation.
> To further verify the consistency between the Ph.D. experts, we conducted an inter-annotator agreement study on 1,000 real infographics.
> For each infographic, we first generated annotations using the model before refinement, then randomly selected one element from those annotations.
> The Ph.D. experts independently corrected the element, adjusting its location and category as needed.
> They were also asked to mark the occurrences of ambiguous chart–HRO boundaries.
> The results on all elements and on 39 elements with ambiguous chart-HRO boundaries are as follows.
> The experts performed consistent corrections (average IoU=0.95, category agreement=0.99).
> These corrections differed from the model predictions, which showed lower agreement (average IoU=0.90, category agreement=0.96), indicating that they can effectively mitigate systematic biases in the model annotations.
> Even for the elements with ambiguous boundaries, the agreement remains high (average IoU=0.84, category agreement=0.92), further confirming the reliability of the corrections.
> We have included this study in **Appendix E**.
>
> |All Elements|Average IoU|Category Agreement|
> |-|-|-|
> |Expert-Expert|**0.95**|**0.99**|
> |Expert-Model|0.90|0.96|
>
> |Elements with Ambiguous Boundaries|Average IoU|Category Agreement|
> |-|-|-|
> |Expert-Expert|**0.84**|**0.92**|
> |Expert-Model|0.77|0.91|
>
> **Systematic error analysis**: We performed systematic error analysis on the 1,250 images used for precision and recall evaluation.
> The errors are summarized as follows.
> The most frequent errors include:
> 1) imprecise bounding-box localization due to ambiguous element boundaries;
> 2) marks or annotations mistakenly detected as HROs due to their high visual similarity; and
> 3) missed tiny elements due to their limited scale.
>
> We have included this analysis in **Appendix F**.
>
> |Error Type|Percentage|
> |-|-|
> |Imprecise bounding-box localization|36.5%|
> |Marks or annotations detected as HROs|32.9%|
> |Missed tiny elements|20.4%|
> |Others|10.2%|

---

> ### Author Response · Authors · 2025-11-25
> **Response to Reviewer fMwe (4/4)**
>
> > **Question 1**: How does annotation quality vary across infographic types (e.g., dense textual vs. image-heavy layouts)?
>
> **A**: Our analysis of remaining annotation errors in **Appendix F** reveals that the most frequent annotation errors are imprecise bounding-box localization due to ambiguous element boundaries (36.5%).
> These ambiguities are more common in image-heavy infographics, where icons and images overlap or lack clear edges.
> In contrast, dense-textual infographics tend to have more regular layouts and text blocks with clear boundaries, which reduces boundary ambiguity.
> As a result, their overall annotation quality is higher.
>
> > **Question 2**: Could the Thinking-with-Boxes approach extend to document reasoning tasks beyond charts (e.g., scientific figures, tables)?
>
> **A**: Yes.
> The core idea behind the Thinking-with-Boxes approach is to provide grounded, element-level annotations that guide step-by-step visual reasoning.
> Since scientific figures and tables also contain semantically meaningful subregions that can be detected and referenced during step-by-step reasoning, the approach naturally extends to other document reasoning tasks.
> While our current work focuses on infographics, extending this approach to other document types is a highly promising research direction for the future.
>
> > **Question 3**: Have the authors considered releasing segmentation masks or relation graphs between detected elements to support compositional reasoning?
>
> **A**: Yes.
> We have now released the segmentation masks for charts, HROs, and chart sub-elements on Hugging Face, extracted using the Segment Anything Model.
> We have introduced the segmentation masks in **Sec. 3.3**.
> We are currently formalizing the definition of relation graphs for infographics and developing methods for their reliable extraction.
> The relation graphs will be released in the final version.
>
> > **Question 4**: Could the authors quantify the annotation process in terms of human effort and computational
>
> **A**: The eight refinement rounds require around $\textdollar$3,000 in human annotation cost and 313 GPU-hours for model fine-tuning on NVIDIA Tesla V100, priced at approximately $\textdollar$0.7 per GPU-hour.
> In total, the refinement process cost roughly $\textdollar$3,220.
> We have added these details in **Appendix E**.
>
> > **Question 5**: What are the computational costs of fine-tuning large detection models (e.g., Co-DETR) on InfoDet relative to standard datasets?
>
> **A**: We have now compared the computational costs for fine-tuning the detection models on InfoDet and standard datasets (PASCAL VOC [6], COCO [7], LVIS [8], and Objects365 [9]) in **Appendix I.2**.
> The results for Co-DETR are as follows.
> Fine-tuning on InfoDet requires more compute than Pascal VOC, is comparable to COCO, and requires less compute than LVIS and Objects365.
>
> |Computational cost (GPU hours)|PASCAL VOC|COCO|LVIS|Objects365|InfoDet|
> |-|-|-|-|-|-|
> |Co-DETR|11|82|113|416|70|
>
> **References**
>
> [1] Leland McInnes, John Healy, and James Melville. Umap: Uniform manifold approximation and projection for dimension reduction. arXiv preprint arXiv:1802.03426, 2018.
>
> [2] Jianhua Lin. Divergence measures based on the Shannon entropy. IEEE Transactions on Information Theory, 37(1):145–151, 2002.
>
> [3] Nadav Borenstein, Karolina Sta´nczak, Thea Rolskov, Natacha Klein Käfer, Natália da Silva Perez, and Isabelle Augenstein. Measuring intersectional biases in historical documents. In Findings of the Association for Computational Linguistics, pp. 2711–2730, 2023.
>
> [4] Enamul Hoque, Parsa Kavehzadeh, and Ahmed Masry. Chart question answering: State of the art and future directions. Computer Graphics Forum, 41(3):555–572, 2022.
>
> [5] Nitesh Methani, Pritha Ganguly, Mitesh M Khapra, and Pratyush Kumar. PlotQA: Reasoning over scientific plots. In Proceedings of the IEEE/CVF Winter Conference on Applications of Computer Vision, pp. 1527–1536, 2020.
>
> [6] Mark Everingham, Luc Van Gool, Christopher K. I. Williams, John M. Winn, and Andrew Zisserman. The PASCAL Visual Object Classes (VOC) challenge. International Journal of Computer Vision, 88(2): 303–338, 2010.
>
> [7] Tsung-Yi Lin, Michael Maire, Serge Belongie, James Hays, Pietro Perona, Deva Ramanan, Piotr Dollár, and C Lawrence Zitnick. Microsoft COCO: Common objects in context. In Proceedings of European Conference on Computer Vision, pp. 740–755, 2014.
>
> [8] Agrim Gupta, Piotr Dollar, and Ross Girshick. LVIS: A dataset for large vocabulary instance segmentation. In Proceedings of the IEEE Conference on Computer Vision and Pattern Recognition, 2019
>
> [9] Shuai Shao, Zeming Li, Tianyuan Zhang, Chao Peng, Gang Yu, Xiangyu Zhang, Jing Li, and Jian Sun. Objects365: A large-scale, high-quality dataset for object detection. In Proceedings of the IEEE/CVF International Conference on Computer Vision, pp. 8430–8439, 2019.

---

### Author Response · Authors · 2025-11-28
**Summary of Reviews and Author Rebuttal**

Dear Reviewers, ACs, and SACs,

We sincerely thank you for your valuable and insightful comments.
We are encouraged by the recognition of our dataset’s scale and novelty, the annotation pipeline's rigor, the Thinking-with-Boxes scheme's effectiveness, and the value of our empirical insights.
We are pleased to find that there is a consensus on the following strengths:

**Scale and novelty of the dataset**: The reviewers highlighted the dataset’s large scale and its contribution in addressing existing dataset gaps.

- R_fMwe: "The scale (over 100k infographics) and fine-grained annotations (charts, HROs, sub-elements) fill a clear research gap."

- R_WrQm: "The proposed InfoDet dataset is both large (≈101K
images, ≈14M boxes) and unusually fine-grained..."

- R_Jyhz: "First large-scale infographic dataset strategically combining real and synthetic data for authenticity and scalability."

- R_Jyhz: "Multi-level annotations: element-level (charts, HROs) and mark-level (26 sub-element categories) providing fine-grained labels."

- R_x562: "This work introduces InfoDet, a large-scale dataset featuring 101k images and 14M annotations, specifically designed to address the complex layouts of infographics..."

**Rigorous and effective model-in-the-loop annotation pipeline**: The reviewers acknowledged both the rigor and effectiveness of the model-in-the-loop annotation pipeline in creating high-quality annotations.

- R_fMwe: "The paper’s methodological rigor and practical significance are outstanding."

- R_fMwe: "The hybrid annotation strategy—combining synthetic and real data under a model-in-the-loop refinement—balances scalability and accuracy, achieving dataset quality comparable to COCO."

- R_Jyhz: "Efficient model-in-the-loop annotation achieving quality comparable to COCO."

**Effectiveness of the Thinking-with-Boxes scheme**: The reviewers recognized that the Thinking-with-Boxes scheme improves VLM performance in infographic understanding.

- R_fMwe: "The grounded CoT prompting demonstrates genuine insight into how structured visual grounding improves reasoning in modern VLMs."

- R_WrQm: "The paper targets a real and under-served pain point in multimodal/chart understanding—current VLMs and chart/infographic QA systems often fail not because they cannot “reason,” but because they cannot reliably ground the relevant regions in cluttered infographic-style inputs."

- R_WrQm: "The “thinking-with-boxes” usage demonstrates that plugging the detected elements into strong VLMs yields consistent gains on challenging, multi-chart settings..."

- R_x562: "The authors provide compelling quantitative evidence for its impact on VLM grounding..."

**Valuable empirical insights**: The reviewers appreciated our empirical analysis that offers valuable insights into model capabilities.

- R_fMwe: "Empirical evaluations are extensive and carefully analyzed, revealing key insights."

- R_Jyhz: "Important empirical findings on model capabilities", including "foundation model limitations", "fine-tuning effectiveness", and "generalization capability".

- R_Jyhz: "Thorough experimental protocol and transparent reporting..."

&nbsp;

We are also truly grateful for the constructive feedback from the reviewers and have carefully addressed their concerns.
Below, we summarize the main review concerns and our responses/solutions.

**Additional quantitative/qualitative analysis**:

- Analyzed the fidelity and bias of the synthetic infographics (R_fMwe).

- Conducted error analysis of the InternImage-based model (R_fMwe).

- Conducted evaluations on more benchmarks, along with statistical analysis, qualitative analysis, and ablation study for the Thinking-with-Boxes scheme (R_fMwe, R_Jyhz, R_x562).

- Developed a hierarchy-guided matching scheme for object detection model training (R_WrQm).

- Evaluated the performance of applying the InternImage-based model to the PosterLayout dataset (R_Jyhz).

**Dataset enrichment**:

- Provided the segmentation masks of charts, HROs, and chart sub-elements (R_fMwe).

- Initiated the development of relation graph extraction (R_fMwe).

- Provided the chart type annotations for the real infographics (R_Jyhz).

**Clarifications**:

- Clarified the expert background, annotation process, and inter-annotator agreement (R_fMwe, R_Jyhz, R_x562).

- Provided additional discussions on the object detection model evaluation results (R_Jyhz).

- Clarified the real-world applications of infographic understanding (R_Jyhz).

- Clarified the necessity of the content text in the Thinking-with-Boxes scheme (R_Jyhz).

- Discussed the limitations in the dataset scope and failure modes (R_x562).

We are confident that these additional analyses, dataset enrichment, and clarifications will effectively address the reviewers' concerns.
We deeply appreciate your careful review, constructive feedback, and valuable time.

---

### Meta-Review · Area_Chair_uKWk · 2026-01-07

**Summary:**

Reviewers generally acknowledge the dataset’s scale, novelty, and sufficient experiments, although one reviewer raised concerns about the limited novelty of the models used. I believe most concerns have been addressed following the rebuttal. Therefore, I recommend accepting this paper.

**Reviewer Scores:**

NA

---

### Decision · Program_Chairs · 2026-01-26

Accept (Poster)